# Beyond Attention Imbalance: Mitigating Hallucinations via Spectral Surgery

**Siqi Lu** [1]  **Wei Suo** [2 3 4]  **Yongbin Zheng** [1]  **Jianhang Yao** [1]  **Wanying Xu** [1]  **Peng Wang** [2 3 4]

## Abstract

While Large Vision-Language Models (LVLMs) achieve remarkable success, hallucinations remain a significant barrier to their reliable deployment. Recent studies primarily attribute these issues to cross-modal attention imbalances; most solutions therefore focus on reweighting visual tokens or suppressing language priors. However, such approaches often overlook the spectral characteristics of the visual information flow and frequently rely on Contrastive Decoding (CD), which doubles inference time. Instead of following conventional approaches, we identify two distinct hallucination patterns—Perceptual-Semantic Dissociation and Localized Fixation—and propose FLASH (**F**requency-**L**ocalized **A**ttention **SH**aping), a training-free and CD-free framework. FLASH utilizes a Spectral Vortex Score to detect vision heads within multi-head attention layers and applies adaptive spectral modulation to rectify the visual information flow during decoding. Empirical results demonstrate that FLASH achieves a superior balance between performance and efficiency compared to SOTA methods.

## 1. Introduction

Large Vision-Language Models (LVLMs) (Liu et al., 2024a; Zhu et al., 2023) extend the functional capabilities of Large Language Models (LLMs), enabling scene understanding and visual reasoning. However, hallucinations remain a formidable challenge (Gunjal et al., 2024; Zhao et al., 2024; Yin et al., 2024), eroding user trust and compromising the reliability of generated content. These errors are es-

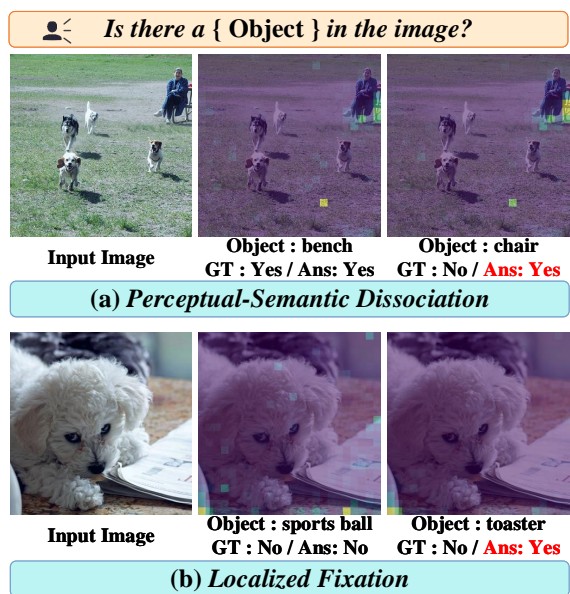

*Figure 1.* Two distinct hallucination patterns. (a) seeing but not understanding and (b) biased attention aggregation. Hallucinated responses are highlighted in red.

pecially problematic in safety-critical applications such as autonomous driving (Jiang et al., 2024; Zhao et al., 2025) and medical diagnostics (Yang et al., 2025; Xia et al., 2024).

Current mitigation studies generally follow two paradigms. The first involves supervised fine-tuning or preference alignment using high-quality datasets (Yu et al., 2024; Rafailov et al., 2024; Sun et al., 2023; Wang et al., 2024a). While effective, these approaches often suffer from the high costs of data curation and model retraining. Another paradigm focuses on post-hoc interventions during the decoding stage (Zheng & Zhang, 2025; Yin et al., 2025; Qian et al., 2026). These methods are increasingly favored for their flexibility.

Unfortunately, most existing training-free methods treat hallucinations merely as an imbalance between visual and textual attention—where dominant language priors override visual evidence (Liu et al., 2024c; Kang et al., 2025; Zhang et al., 2025). Consequently, research has focused heavily on manipulating attention weights in the spatial domain, largely overlooking the intrinsic spectral characteristics of visual information. Furthermore, the prevailing reliance on Contrastive Decoding (CD) (O'Brien & Lewis, 2023; Leng

[1]College of Intelligence Science and Technology, National University of Defense Technology, China [2]School of Computer Science, Northwestern Polytechnical University, China [3]Ningbo Institute, Northwestern Polytechnical University, China [4]National Engineering Laboratory for Integrated Aero-Space-Ground-Ocean, China. Correspondence to: Yongbin Zheng <zyb-nudt@nudt.edu.cn>.

*Proceedings of the 43rd International Conference on Machine Learning*, Seoul, South Korea. PMLR 306, 2026. Copyright 2026 by the author(s).

et al., 2023; Wang et al., 2024b) significantly increases the computational burden, hindering real-time deployment.

These issues motivate our study to address three questions: **(1)** Are hallucinations merely a byproduct of simplistic attention imbalances, or do they arise from more intricate patterns? **(2)** Can the frequency-domain perspective serve as a potent lens for both understanding the underlying mechanisms of hallucinations and facilitating their mitigation? **(3)** Can we design an effective mitigation framework that bypasses the computational overhead inherent in CD?

In this paper[1], we demonstrate through statistical analysis that hallucinations are not merely a symptom of attention imbalance. Instead, they manifest in two distinct modes: **Perceptual-Semantic Dissociation** (PSD) and **Localized Fixation** (LF) (Fig. 1)—each interpretable from a frequency-domain perspective. Our findings reveal that simply amplifying spatial attention often yields marginal gains, with performance remaining heavily dependent on the expensive CD (Fig. 2). Motivated by these insights, we shift our focus to the frequency domain and propose FLASH (**F**requency-**L**ocalized **A**ttention **SH**aping). FLASH is a training-free framework that introduces the **S**pectral **V**ortex **S**core (SVS) to identify vision heads within Multi-Head Attention (MHA) layers. By applying adaptive spectral modulation to the value matrices and attention scores, FLASH rectifies distorted signals, encouraging the model to maintain granular focus on localized regions while enhancing attention to the global context. This mechanism effectively mitigates hallucinations arising from both PSD and LF. Extensive evaluations across diverse benchmarks confirm that FLASH achieves a SOTA balance between hallucination mitigation and inference efficiency.

In summary, our main contributions are as follows:

1) We demonstrate that hallucinations in LVLMs extend beyond simple attention imbalances, manifesting as two distinct patterns: PSD and LF. Furthermore, our analysis shows that although current methods aim to optimize attention distribution during the decoding phase, they heavily rely on CD, which inevitably increases inference cost.

2) We introduce a spectral perspective to both analyze the etiology of hallucinations and design a targeted solution. To this end, we propose FLASH. To the best of our knowledge, this is the first work to leverage spectral analysis as a core mechanism for analyzing and mitigating hallucinations in LVLMs.

3) We validate FLASH across diverse benchmarks, demonstrating that it surpasses or matches SOTA CD-based methods. By bypassing CD, FLASH provides a more efficient

---

[1]Our code is available at https://github.com/a6103121/FLASH.

and scalable solution for mitigating hallucinations.

**Conflict of Interest Disclosure.** There are no conflicts of interest in our work.

## 2. Preliminary

A typical LVLM architecture consists of three core components: a visual encoder, a modality projection layer, and an LLM backbone (Liu et al., 2024a; Zheng & Zhang, 2025). Both the visual encoder and the LLM are typically Transformer-based. The projection layer acts as a bridge, mapping encoded visual features into the text embedding space to ensure cross-modal alignment. These visual tokens are then concatenated with textual tokens to form a unified input sequence, from which the LLM generates responses in an autoregressive manner.

A standard LLM decoder consists of $L$ stacked MHA layers, each containing $H$ attention heads. For the $h$-th head in the $l$-th layer, the attention mechanism is defined as:

$$S_{l,h}(Q_{l,h}, K_{l,h}) = \frac{Q_{l,h}K_{l,h}^T}{\sqrt{d_k}}, \tag{1}$$

$$A_{l,h}(Q_{l,h}, K_{l,h}, V_{l,h}) = \text{softmax}(S_{l,h}) \cdot V_{l,h}, \tag{2}$$

where $Q$, $K$, $V \in \mathbb{R}^{n \times d_k}$ represent the query, key, and value matrices, respectively; $d_k$ is the head dimension, and $n$ denotes the sequence length.

The outputs from all heads are concatenated and transformed via a linear projection to produce the layer's final representation. After $L$ successive blocks, the model yields the final layer hidden state $h_L$. During generation of the $s$-th token, the hidden state $h_{L,s}$ is mapped to the vocabulary logit space through a linear layer $\phi$, yielding the conditional probability distribution:

$$y_t \sim \text{softmax}(h_{L,s} \cdot \phi). \tag{3}$$

## 3. Spectral Diagnosis of Hallucinations

### 3.1. Revisiting Attention Imbalance

Recent studies have identified a striking disparity in LVLMs: despite visual tokens numerically dominating the input, the attention weights allocated to the visual modality remain low (Kang et al., 2025; Liu et al., 2024c). This imbalance is widely believed to induce an over-reliance on language priors, causing the model to neglect grounded visual evidence. Leading approaches, such as PAI (Liu et al., 2024c), attempt to counteract this by amplifying visual attention scores to create a vision-augmented branch and subsequently employ CD to prioritize grounded logits. However, beyond the efficiency penalties arising from CD-induced inference overhead, it remains to be determined whether rescaling visual weights yields robust, independent improvements or

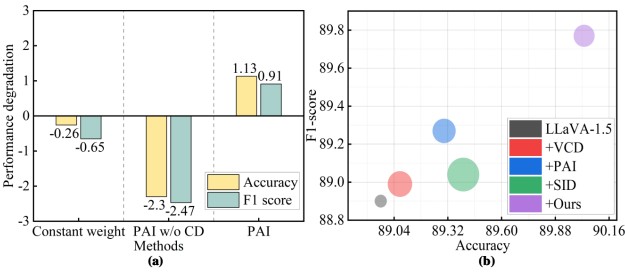
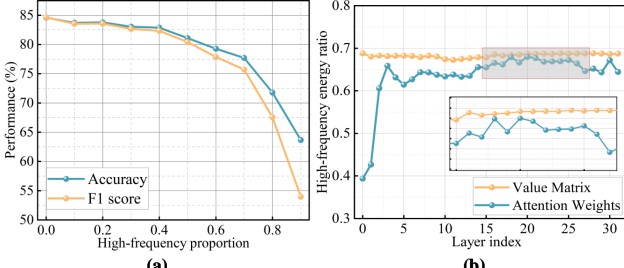

*Figure 2.* Comparison of different methods for mitigating hallucinations. Results are reported relative to Vanilla LLaVA-1.5 on the POPE (COCO-R) dataset. (a) For fixed-weight configurations, a gain coefficient of 1.5 is applied to visual tokens. (b) Circle size indicates the inference time.

*Figure 3.* Impact of low-pass filtering on performance and spectral energy distribution. (a) Impact of different frequency band information on model performance. (b) High-frequency energy ratio of the Value Matrix and Attention Weights across layers, with the high-frequency cutoff fixed at 0.5. We employ random sampling in this experiment.

whether its perceived efficacy is merely an artifact of the CD process.

To respond to these questions, we evaluated discriminative performance on the POPE benchmark (Li et al., 2023b) under several configurations: the vanilla baseline, the vanilla PAI, PAI without CD (w/o CD), and PAI with fixed linear gains applied to visual tokens.

As shown in Fig. 2, our results reveal that linear amplification of visual attention in the spatial domain is insufficient for robust hallucination mitigation and can even be counterproductive. The observed performance gains of existing methods are primarily attributable to the CD rather than to the attention manipulation itself. Our results suggest that achieving low-cost mitigation requires shifting from coarse-grained scaling to a more in-depth analysis of information flow within the visual token, thereby bypassing the overhead of CD.

### 3.2. Deconstructing the Hallucination Patterns

To elucidate the manifestations of hallucinations at the visual token level, we conducted diagnostic visualizations on POPE. By extracting attention weights from the 16th MHA layer and reshaping the visual tokens into 2D spatial maps (Liu et al., 2025), we performed statistical experiments comparing the attention distributions of grounded responses versus hallucination responses. The results reveal two distinct hallucination patterns in LVLMs:

**(1) Perceptual-Semantic Dissociation (PSD).** This pattern signifies a rupture between spatial localization and semantic interpretation. The model successfully "anchors" its attention to the correct visual region but fails to execute fine-grained semantic parsing. As shown in Fig. 1a, when queried about the "bench" and the "chair," the model's attention consistently converges on the exact same spatial coordinates (the ground-truth location of the bench). This suggests that although the model possesses the perceptual capacity to localize regions based on textual prompts, it lacks the

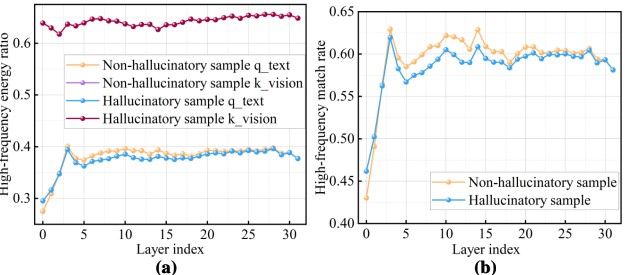

*Figure 4.* High-frequency (HF) energy analysis of queries and keys. (a) Comparison of HF energy distributions between hallucinatory and non-hallucinatory samples. (b) HF energy match rate, defined as the ratio HF($q_{\text{text}}$) / HF($k_{\text{vision}}$).

discriminative resolution to distinguish fine-grained features (e.g., the elongated form of a bench), resulting in a semantic best guess rather than grounded reasoning.

**(2) Localized Fixation (LF).** This pattern manifests as a pathological capture of attention by irrelevant visual locations. As shown in Fig. 1b, the model frequently becomes trapped within contiguous regions that offer marginal semantic utility for the current token generation. Although similar to the "attention sink" (Xiao et al., 2024; Kang et al., 2025), LF is distinguished by its intense spatial contiguousness—attention is concentrated in dense, localized patches rather than distributed across sparse outliers. This indicates that the model is not merely ignoring visual tokens but is actively distracted by localized noise, which obscures its contextual observational perspective and induces a drift toward hallucinated responses.

Consequently, an indiscriminate "one-size-fits-all" enhancement of visual tokens is inadequate for addressing the diverse modes of hallucinations. It is necessary to establish a unified framework to tackle different hallucination patterns.

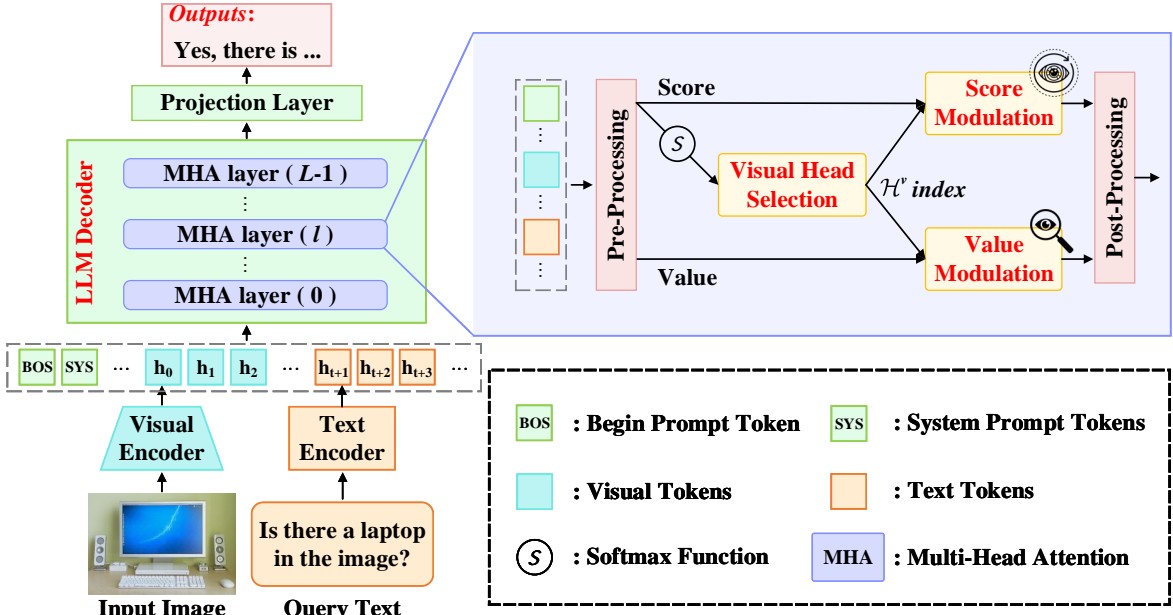

*Figure 5.* Overview of the FLASH framework. FLASH rectifies hallucinatory signals via spectrum modulation of attention scores and value matrices within the MHA layers. Specifically, spectral shaping of scores is designed to broaden contextual coverage (**seeing comprehensively**), while modulation of value matrices aims to enhance perceptual fidelity (**seeing clearly**).

### 3.3. Spectral Analysis of Information Flow within MHA

**Spectral components serve distinct roles.** To quantify the impact of spectral composition on model performance, we performed spectral perturbation analysis by modulating the cutoff frequency of a low-pass filter applied to the hidden states. As shown in Fig. 3a, the model exhibits nonlinear sensitivity to filter cutoff: performance remains stable when high-frequency components remain below 0.4, begin to decline at 0.5, and collapse by 0.8. This suggests that low-frequency components underpin the model's baseline reliability, whereas high-frequency components govern its performance bottlenecks.

To design precise interventions for diverse hallucination patterns, we dissect two critical components of MHA: the value matrix, which encodes the visual content, and the interaction between text queries ($Q^t$) and visual keys ($K^v$), which determines spatial attention priority. Our spectral analysis yields the following insights:

**The MHA filters out high-frequency information from visual tokens.** Although the low-pass filtering nature of MHA in Transformers is well-documented (Park & Kim, 2022; Dong et al., 2023), its relationship with hallucinations still requires further exploration. We conducted a comparative analysis of high-frequency energy proportions between the value matrix and the attention output. As shown in Fig. 3b, a gap persists between the raw visual features and their aggregated representations across all layers. This observation experimentally confirms that MHA induces an

irreversible smoothing of fine-grained visual details during cross-modal alignment. Although the mutations in attention output in deeper layers suggest a compensatory attempt to capture localized high-frequency structures, the energy ratio of the output consistently fails to recover the original richness of the value matrix. This indicates that hallucinations arise from the attention mechanism's inability to propagate the precise visual evidence required for reasoning, which directly corroborates our observed PSD hallucinations.

**Spectral misalignment between $Q^t$ and $K^v$ serves as the mechanistic driver of LF.** Fig. 4a shows that while $K^v$ maintains stable high-frequency energy, hallucinatory $Q^t$ exhibits a significant high-frequency deficit in mid-to-late layers, resulting in a suppressed matching rate after Layer 3 (Fig. 4b). This indicates that sharp, high-frequency $K^v$ act as deceptive interference sources. When $Q^t$ lack the spectral precision to align with correct anchors, attention is easily captured by these local noise anchors, leading to over-fixation on incorrect tokens (e.g., "toaster" in Fig. 1b).

## 4. Proposed Methods

### 4.1. Overview

Fig. 5 illustrates the overview of FLASH. Designed to operate within the decoder, FLASH adaptively modulates the spectral components of both attention scores and value matrices to enable global context integration while preserving high-fidelity visual perception. FLASH first selects vision heads within the MHA layers to ensure precise intervention.

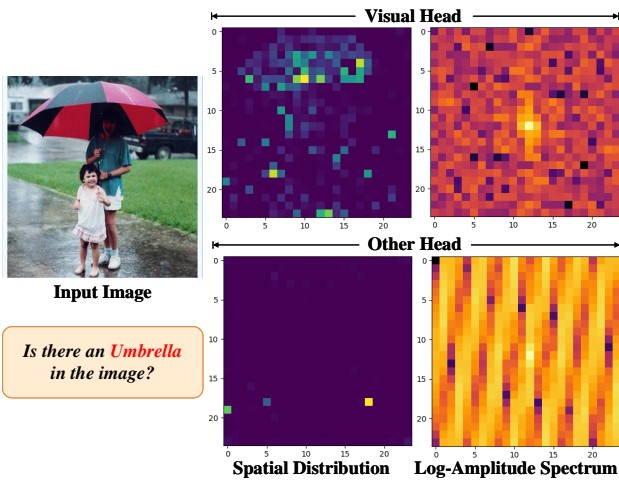

*Figure 6.* Spatial distributions and log-magnitude spectra of vision heads compared with those of other heads.

To counteract the hallucination patterns described in Sec. 3.2, FLASH introduces a dual-stream spectral modulation mechanism: shaping the spectral representation of attention scores to foster comprehensive contextual awareness, and refining the spectral representation of value matrices to amplify object-relevant tokens. FLASH achieves these enhancements through a single inference pass.

### 4.2. Spectrum-Based Vision Head Selection

In the MHA layers of LVLMs, individual heads exhibit functional heterogeneity (Zheng et al., 2024; Zhang et al., 2024). Applying modulation indiscriminately across all heads is suboptimal, as it may compromise linguistic coherence in the generated content and introduce additional computational costs. In FLASH, we distinguish vision-related heads ($\mathcal{H}^v$) from the remaining ($\mathcal{H}^o$) by characterizing the spectral signatures of their attention weights.

**Qualitative Analysis.** We performed a qualitative analysis to characterize the spectral discrepancies between $\mathcal{H}^v$ and $\mathcal{H}^o$. Reconstructing and profiling the attention weight spectra reveals pronounced disparities between $\mathcal{H}^v$ and $\mathcal{H}^o$ (Fig. 6). These observations lead to two key insights:

**(1) $\mathcal{H}^v$ exhibits higher spectral energy concentration.** Recent studies (Kang et al., 2025) have identified "attention sinks" within visual tokens, analogous to those observed in textual sequences. These sinks are markedly more pronounced in $\mathcal{H}^o$, where they manifest spatially as sparse, high-magnitude singularities resembling Dirac delta functions. From a signal processing perspective, such impulse-like signals possess broadband characteristics, causing their spectral energy to disperse across the entire frequency domain rather than concentrate within the low-frequency components. Consequently, $\mathcal{H}^o$ manifests a more diffuse energy

distribution compared to $\mathcal{H}^v$.

**(2) $\mathcal{H}^v$ displays structured vortex-like spectra, whereas $\mathcal{H}^o$ exhibits periodic ripples.** In the spatial domain, the dominance of sparse sink tokens in $\mathcal{H}^o$ introduces multiple localized peaks, which generate periodic ripples across the spectrum. In contrast, $\mathcal{H}^v$ maintains consistent activation across spatially contiguous semantic regions. This continuous spatial activation suppresses the spectral dominance of individual sink tokens. Due to the inherent conjugate symmetry of the FFT, the spectra of $\mathcal{H}^v$ manifest as vortex-like structures.

**Quantitative Assessment.** Leveraging these qualitative insights, we introduce the Spectral Vortex Score (SVS), a composite metric designed to quantitatively identify $\mathcal{H}^v$.

Given the attention weights for reshaped visual tokens $A_{l,h}^v \in \mathbb{R}^{\sqrt{N} \times \sqrt{N}}$ for the $h$-th head in layer $l$ (where $N$ is the visual token length), we project them into the frequency domain via FFT and compute the log-magnitude spectrum $M$:

$$M(p,q) = 20 \log_{10}(|\mathcal{F}(A(x,y))| + \epsilon), \quad (4)$$

where $\mathcal{F}$ denotes the FFT operator and $\epsilon$ is a small positive constant. We omit subscripts for brevity. Our SVS consists of two components:

**Spectral Energy Concentration.** According to our first observation, $\mathcal{H}^v$ captures structured semantic information, which manifests as energy concentration in the central low-frequency region. We define a low-frequency mask $\mathcal{M}_L$ with radius $R$ (in this paper, we set $R = 1$):

$$\mathcal{M}_L(p,q) = \begin{cases} 1, & \text{if } \sqrt{(p-p_c)^2 + (q-q_c)^2} \leq R \\ 0, & \text{otherwise} \end{cases} \quad (5)$$

where $(p_c, q_c)$ denotes the pixel coordinates of the spectral DC. The concentration score $\mathcal{L}_r$ can be defined as:

$$\mathcal{L}_r = \frac{\sum_{p,q} M(p,q) \cdot \mathcal{M}_L(p,q)}{\sum_{p,q} M(p,q)}. \quad (6)$$

**Spectral Isotropy.** Building on our second observation, $\mathcal{H}^o$ exhibits highly directional ripples due to periodic impulses from attention sinks, whereas $\mathcal{H}^v$ displays isotropic, vortex-like structures. We employ orientation coherence to quantify this structural disparity. Computing the gradient field of the magnitude spectrum $\nabla M = (G_p, G_q)$, we further obtain:

$$G_{mag} = \sqrt{G_p^2 + G_q^2}, \qquad \theta = \text{atan2}(G_q, G_p). \quad (7)$$

We formulate the weighted orientation coherence to measure the strength of the dominant orientation:

$$\mathcal{C} = \frac{\sqrt{\left(\sum G_{mag} \cos(2\theta)\right)^2 + \left(\sum G_{mag} \sin(2\theta)\right)^2}}{\sum G_{mag}}, \quad (8)$$

where $\mathcal{C} \to 1$ indicates a highly anisotropic spectrum. We thus define the spectral isotropy as $\mathcal{O}_c = 1 - \mathcal{C}$.

The SVS is the sum of these normalized components:

$$\text{SVS} = \mathcal{L}_r + \mathcal{O}_c, \qquad (9)$$

where a higher SVS indicates a larger probability of $\mathcal{H}^v$. Inspired by (Kang et al., 2025), we first exclude heads with an aggregate attention weight below threshold $\tau$, and then identify the top $k$ heads with the highest SVS values as $\mathcal{H}^v$.

### 4.3. Dual-stream Spectrum Modulation

To mitigate the diverse patterns of object hallucinations, we introduce Dual-stream Spectrum Modulation (DSM), a strategy designed to adaptively modulate the information flow of visual tokens in the decoder. Diverging from spatial-domain reweighting (Liu et al., 2024c; Zhang et al., 2025), DSM performs adaptive redistribution of spectral energy within both the attention scores and value matrices. By operating in the spectral domain, DSM circumvents the heavy computational overhead of CD and mitigates noise-induced interference common in spatial manipulations. Furthermore, this approach enhances the interpretability of the model's internal representations.

DSM comprises two parallel branches—the V-stream and the S-stream—which respectively modulate the value matrices ($V$) and the attention scores ($S$) within MHA layers. We begin by employing the Discrete Cosine Transform (DCT) for frequency-domain projection, leveraging its superior energy harvesting capabilities. For a spatially reshaped feature map $X^m \in \mathbb{R}^{\sqrt{N} \times \sqrt{N}}$ (where $m \in \{V, S\}$), the spectral coefficients $C(p, q)$ are computed as:

$$C^m(p, q) = \mathcal{D}(X^m), \qquad (10)$$

where $\mathcal{D}(\cdot)$ denotes the DCT operator; $C^m(0, 0)$ is the DC component, while coefficients with higher indices $(p, q)$ represent higher spatial frequencies.

We use a radial distance matrix $D_{p,q}$ to characterize the spectral distribution, which will be employed in our DSM:

$$D_{p,q} = \sqrt{p^2 + q^2}, \quad p, q \in \{0, 1, \dots, \sqrt{N} - 1\}. \quad (11)$$

**V-stream**. PSD occurs when a model identifies relevant regions based on queries but fails to capture fine-grained details. This suggests that although the model leverages low-frequency components for localization, its sensitivity to granular semantic features is limited. Prior research (Wei et al., 2021; Wang et al., 2020; Xu et al., 2020) indicates that such fine-grained information is predominantly embedded in high-frequency components. Unfortunately, our findings in Sec. 3.3 reveal that the inherent low-pass filtering property

of MHA attenuates these high-frequency cues in the value matrix, thereby weakening the model's perceptual acuity.

By enhancing the high-frequency energy of the value matrix through adaptive spectral modulation, we can counteract this collapse. We first construct a soft mask $\mathcal{M}_v$:

$$\mathcal{M}_v(p, q) = 1 + \lambda_v \cdot \text{sigmoid}(D_{p,q} - \alpha\sqrt{N}), \qquad (12)$$

where $\lambda_v$ denotes the modulation strength and $\alpha$ is the frequency cutoff threshold. We fix $\mathcal{M}_v(0, 0) = 1$ to preserve the DC component. We perform the intervention in the log-magnitude domain—a decision predicated on the power-law distribution of visual features (Field & David, 1987; Simoncelli & Olshausen, 2001):

$$\tilde{C}^V = \text{sgn}(C^V) \cdot e^{\log |C^V| \cdot \mathcal{M}_v}, \qquad (13)$$

where $\text{sgn}(\cdot)$ is the sign function, ensuring correctness of all DCT coefficients. We further discuss the advantages of logarithmic modulation in **Appendix A.3** and **A.5**.

The weighted spectrum is mapped back to the spatial domain using the IDCT, followed by an adaptive energy modulation step to calibrate the resulting features:

$$\tilde{V} = \text{IDCT}(\tilde{C}^V), \qquad (14)$$

$$V^* = \tilde{V} \cdot \frac{||V||_F}{||\tilde{V}||_F + \epsilon}. \qquad (15)$$

The naive scaling of distinct frequency bands poses a substantial risk of destabilizing the joint distribution of multi-modal tokens, often necessitating delicate hyperparameter tuning. In contrast, our proposed modulation—leveraging the synergy between Eq. 13 and Eq. 15—acts as a spectral energy reallocation mechanism. According to Parseval's Theorem [2], enforcing a constant F-norm constraint ensures that amplification of high-frequency components inherently squeezes redundant energy from low-frequency bands and reallocates it to suppressed fine-grained details. This intrinsic zero-sum energy dynamic enables the model to refocus on fine-grained semantic features without shifting global feature magnitudes or compromising pre-trained multimodal alignment. Please refer to **Appendix A.3** for theoretical details. Consequently, this modulation provides a robust, plug-and-play solution for mitigating PSD while preserving numerical stability in the representation space.

**S-stream** is specifically designed to counteract LF, a phenomenon characterized by pathological over-concentration of spatial attention on irrelevant visual regions. Since attention scores govern the allocation of visual priorities in LVLMs, we refine their spectral energy distribution to enhance global contextual awareness and prevent the model from being distracted by localized noise.

---

[2]Parseval's Theorem: The total energy of a signal in the time domain equals that in the frequency domain.

Similar to Eq. 12, we construct a spectral soft mask $\mathcal{M}_s$ to modulate the S-stream:

$$\mathcal{M}_s(p, q) = (1 - \lambda_s) + \lambda_s \cdot \text{sigmoid}(\alpha\sqrt{N} - D_{p,q}), \quad (16)$$

where $\lambda_s$ denotes the modulation strength and $\alpha$ is the frequency cutoff threshold. We also strictly enforce $\mathcal{M}_s(0, 0) = 1$ to preserve the DC component.

In contrast to the V-stream, the S-stream operates directly in the linear spectral domain. This design prevents excessive distortion of the attention distribution during the subsequent softmax, which could otherwise destabilize model convergence. The weighted spectral coefficients are obtained via the Hadamard product: $\tilde{C}^S = C^S \odot \mathcal{M}_s$.

The weighted coefficients are then projected back into the spatial domain via the IDCT, followed by norm calibration:

$$\tilde{S} = \text{IDCT}(\tilde{C}^S) \quad (17)$$

$$S^* = \tilde{S} \cdot \frac{||S||_F}{||\tilde{S}||_F + \epsilon} \quad (18)$$

Similar to V-stream, the coordination of Eq. 16 and Eq. 18 adaptively "pushes" energy suppressed in high-frequency bands toward low-frequency regions, thereby achieving a compensatory redistribution of spectral energy. In summary, the S-stream acts as a spectral regularizer that immunizes the model against localized artifacts and facilitates more comprehensive perception of visual tokens.

Finally, the attention output $A^*$ is calculated from the modulated attention score $S^*$ and the value matrix $V^*$:

$$A^* = \text{softmax}(S^*) \cdot V^*, \quad (19)$$

## 5. Experiment

### 5.1. Experimental Setting

**Datasets and Evaluation Metrics.** We conducted experiments on both discriminative and generative tasks to demonstrate the effectiveness of FLASH. Following previous work, we use POPE (Li et al., 2023b) and MME (Fu et al., 2025) for discriminative benchmarks, while CHAIR (Rohrbach et al., 2019) and AMBER (Wang et al., 2023) are used for generative evaluation. Performance on discriminative tasks is quantified using Accuracy, F1-score, and Score. For generative tasks, we report CHAIR$_S$, CHAIR$_I$, Cover, Hal, and Cog (Wang et al., 2023) scores to measure hallucination tendencies. Please refer to **Appendix A.8.1** for further details.

**Baselines.** We selected two representative families of LVLMs as base models: LLaVA-1.5 (Liu et al., 2024a) and Shikra (Chen et al., 2023a). We employed the 7B and 13B variants of LLaVA-1.5 to assess the adaptability across different model scales. To demonstrate the performance and efficiency of FLASH, we compared it with three SOTA methods: VCD (Leng et al., 2023), PAI (Liu et al., 2024c), and SID (Huo et al., 2025). Further details can be found in **Appendix A.8.2**.

**Implementation Details.** All experiments were conducted on two NVIDIA RTX 4090 GPUs. Following the analysis in Fig. 3a, the high-frequency cutoff was set to 0.5 across all LVLMs. To accommodate the diverse visual token lengths, decoder architectures of different models, and task requirements, we adjusted the modulation strengths $\lambda_v$ and $\lambda_s$, as well as the vision head selection parameters $k$ and $\tau$, for each specific LVLM. Detailed configurations are provided in **Appendix A.8.3**.

### 5.2. Quantitative Evaluation

Table 1 summarizes the performance across multiple benchmarks, where FLASH outperforms other methods on most metrics. FLASH demonstrates more significant gains on discriminative benchmarks than on generative tasks, highlighting its robust spatial awareness. Although it is suboptimal on a few generative metrics, FLASH maintains overall competitive performance. Additionally, the inference efficiency of different methods is detailed in **Appendix A.6**. Detailed task-level scores for MME and qualitative visualizations of the generative task are provided in **Appendix A.5** and **Appendix A.7**, respectively. In summary, these results indicate that FLASH achieves a better balance between performance and efficiency in mitigating hallucinations.

### 5.3. Ablation Study

In this section, we report the primary ablation study results. Please refer to the **Appendix A.5** for additional experiments.

The ablation results in Table 2 demonstrate that each component is indispensable to the model's overall performance. Specifically, visual head selection enables targeted modulation, thereby preserving the integrity of language heads. Furthermore, the two streams within the DSM yield independent performance gains, emphasizing their specialized roles in mitigating distinct hallucination patterns. Ultimately, the integrated DSM operates synergistically to tackle diverse modes, achieving the most robust performance.

## 6. Related Work

**Data-Driven Instruction Tuning** methods focus on improving data quality and using specialized instruction tuning to better anchor models in visual evidence. Early research suggested that hallucinations often stem from a mismatch between visual features and the model's language priors. LLaVA-1.5 (Liu et al., 2024a) demonstrated that high-quality data coupled with a simplified linear projector

*Table 1.* Comparison with SOTA methods. Rows shaded in **green** denote the performance of vanilla LVLMs, while subsequent rows report results after integrating various hallucination mitigation methods into these models. The evaluation spans both discriminative benchmarks (POPE and MME) and generative benchmarks (CHAIR and AMBER). For each LVLM, the best and second-best results among the mitigation methods are highlighted in **pink** and **purple**, respectively. All experiments employed greedy decoding.

| LVLMs | POPE-MS-COCO | | | | | | MME | CHAIR | | AMBER | | |
| | Random | | Popular | | Adversarial | | | | | | | |
| | Acc. ↑ | F1 ↑ | Acc. | F1 | Acc. | F1 | Score ↑ | CHAIR$_S$ ↓ | CHAIR$_I$ ↓ | Cover ↑ | Hal ↓ | Cog ↓ |
|---|---|---|---|---|---|---|---|---|---|---|---|---|
| LLaVA-1.5 7B | 88.97 | 88.90 | 85.63 | 86.03 | 79.23 | 80.99 | 621.67 | 48.10 | 12.75 | 51.00 | 30.50 | 3.20 |
| +VCD | 89.07 | 88.99 | 85.60 | 85.99 | 79.27 | 81.00 | 636.67 | 48.80 | 12.85 | 51.55 | 27.00 | 2.85 |
| +PAI | 89.30 | 89.27 | 86.07 | 86.45 | 79.23 | 81.06 | 636.67 | 47.80 | 12.35 | 47.10 | 24.25 | 1.45 |
| +SID | 89.40 | 89.04 | 85.93 | 85.93 | 80.33 | 81.38 | 606.67 | 48.10 | 12.40 | 52.40 | 30.50 | 2.15 |
| +Ours | 90.03 | 89.77 | 86.53 | 86.62 | 80.53 | 81.75 | 636.67 | 47.50 | 12.55 | 51.00 | 25.00 | 2.15 |
| Shikra | 84.67 | 84.58 | 82.10 | 82.43 | 77.93 | 79.20 | 458.33 | 49.20 | 13.80 | 52.75 | 42.00 | 3.65 |
| +VCD | 84.67 | 84.51 | 83.03 | 83.12 | 78.30 | 79.38 | 463.33 | 47.70 | 13.70 | 52.35 | 38.75 | 3.45 |
| +PAI | 85.20 | 84.37 | 82.23 | 81.78 | 78.93 | 79.10 | 461.67 | 50.40 | 14.10 | 51.65 | 38.25 | 3.20 |
| +SID | 83.27 | 82.37 | 82.83 | 81.99 | 78.33 | 78.29 | 473.33 | 50.60 | 13.80 | 53.20 | 35.75 | 2.60 |
| +Ours | 86.03 | 84.99 | 83.30 | 82.56 | 79.80 | 79.65 | 468.33 | 49.60 | 14.05 | 52.65 | 35.75 | 3.10 |
| LLaVA-1.5 13B | 89.73 | 89.99 | 85.80 | 86.66 | 80.47 | 82.53 | 598.33 | 42.80 | 11.90 | 50.30 | 26.50 | 2.35 |
| +VCD | 89.77 | 90.00 | 85.73 | 86.59 | 80.57 | 82.58 | 606.67 | 42.20 | 12.00 | 50.50 | 26.25 | 2.55 |
| +PAI | 90.03 | 90.20 | 86.17 | 86.90 | 81.07 | 82.89 | 598.33 | 37.60 | 10.50 | 49.50 | 28.50 | 1.70 |
| +SID | 90.17 | 89.80 | 83.60 | 84.06 | 80.80 | 81.83 | 596.67 | 41.20 | 10.90 | 49.60 | 26.00 | 2.70 |
| +Ours | 90.60 | 90.71 | 86.70 | 87.35 | 80.80 | 82.70 | 601.67 | 42.70 | 11.75 | 50.45 | 23.00 | 2.25 |

*Table 2.* Comparison of ablation results. "Select" denotes the visual head selection mechanism. † denotes using only the modulation strategy corresponding to that stream. We report the average results across the three splits of POPE-COCO.

| Methods | POPE | | AMBER | | |
| | Acc.↑ | F1↑ | Cover↑ | Hal↓ | Cog↓ |
|---|---|---|---|---|---|
| Vanilla | 84.61 | 85.31 | 51.00 | 30.50 | 3.20 |
| w/o Select | 85.69 | 85.89 | 51.85 | 25.25 | 2.20 |
| w/ Select | 85.70 | 86.05 | 51.00 | 25.00 | 2.15 |
| S-stream† | 85.38 | 85.73 | 50.50 | 25.25 | 2.30 |
| V-stream† | 85.55 | 85.91 | 51.15 | 25.50 | 2.20 |

is more effective than dataset size in mitigating hallucinations. ShareGPT4V (Chen et al., 2023b) extended this by using detailed image descriptions generated by proprietary models to sharpen the model's perception of fine-grained attributes. To address "over-generalization," RLHF-V (Yu et al., 2024) introduced segment-level corrective feedback via reinforcement learning, whereas Silkie (Li et al., 2023a) and LLaVA-RLHF (Sun et al., 2023) adopted Direct Preference Optimization (Rafailov et al., 2024) to encourage factual consistency. More recently, Bunny (He et al., 2024) and LLaVA-NeXT (Li et al., 2024; Liu et al., 2024b) identified low spatial resolution as a key driver of errors in small-object recognition, leading them to advocate for higher-resolution visual inputs. Although these methods yield significant improvements, they remain constrained by the high cost of data collection, complex training pipelines, and heavy computational requirements.

**Training-Free Decoding Intervention.** Prior research (Liu et al., 2024c; Leng et al., 2023) indicates that hallucinations in LVLMs during autoregressive generation primarily stem from an over-reliance on linguistic priors, which hinders the model's ability to ground its outputs in actual visual evidence. To bypass the costs of retraining, the research community has increasingly turned to plug-and-play interventions applied during the decoding phase. Early schemes focused on analyzing spatial attention distributions; for instance, OPERA (Huang et al., 2024) identified the over-trust pitfall where models fixate on specific tokens. Building on CD, VCD (Leng et al., 2023) introduced Gaussian noise to construct contrastive samples that counteract linguistic bias. This paradigm has inspired numerous variants: PAI (Liu et al., 2024c) amplifies visual grounding by modulating attention scores; DoLA (Chuang et al., 2024) and ICD (Wang et al., 2024b) mitigate signal submergence through layer-wise contrast or instruction-based bias calibration. Similarly, M3ID (Favero et al., 2024), IBD (Zhu et al., 2024), and Octopus (Suo et al., 2025) have refined the CD framework through multi-layer feature contrast and strategic integration.

Despite performing well, CD-based methods are inherently limited by a doubling of inference latency. Furthermore, the introduction of contrastive noise can occasionally compromise generative quality (Huo et al., 2025). Recent efficient alternatives like VAF (Yin et al., 2025), ADAPTVIS (Chen et al., 2025), and VAR (Kang et al., 2025) have begun exploring internal mechanisms—such as energy distribution or logit calibration—to bypass the dual-stream architecture.

In this paper, we depart from conventional spatial-domain manipulations by pioneering a spectral analysis of hallucination dynamics.

## 7. Conclusion & Discussion

In this paper, we demonstrate that the effectiveness of recent post-hoc hallucination mitigation methods heavily relies on CD strategies, which incur a substantial computational penalty. By analyzing the spatial attention representations of visual tokens and their spectral features, we refine the types of hallucinations into PSD and LF. To address these, we introduce FLASH, which is a training-free framework that mitigates hallucinations from a spectral perspective. FLASH incorporates two core components: vision head selection and dual-stream adaptive spectral modulation. Experimental results confirm that FLASH outperforms SOTA methods in balancing performance and efficiency, successfully bypassing the standard CD paradigm through targeted processing of distinct hallucinatory patterns.

Despite its efficacy, FLASH entails certain technical limitations. First, although it circumvents the CD paradigm and significantly reduces computational complexity, FLASH still relies on multiple spectral transformations and filtering operations. Second, future research may enable targeted modulation of hallucinations by identifying their underlying patterns—thereby further minimizing inference latency. Third, FLASH simultaneously applies value modulation and attention modulation; decoupling or selectively applying these modulations could yield more targeted mitigation strategies while accelerating overall processing. Finally, due to FLASH's analytical design principles, it remains unclear whether it is applicable to Q-Former-based models. We will prioritize these directions in our future work.

## Impact Statement

This paper presents work whose goal is to advance the field of Machine Learning. There are many potential societal consequences of our work, none which we feel must be specifically highlighted here.

## Acknowledgment

This work was supported in part by the National Natural Science Foundation of China under Grant 62273353.

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

# A. Appendix

## A.1. Visualization of Different Hallucination Patterns

In Section 3.2, we propose a refined taxonomy of hallucinations, namely Perceptual-Semantic Dissociation (PSD) and Localized Fixation (LF). Owing to space constraints in the main text, only one representative sample per category was presented. In this section, we provide a broader range of qualitative results to offer more robust evidence.

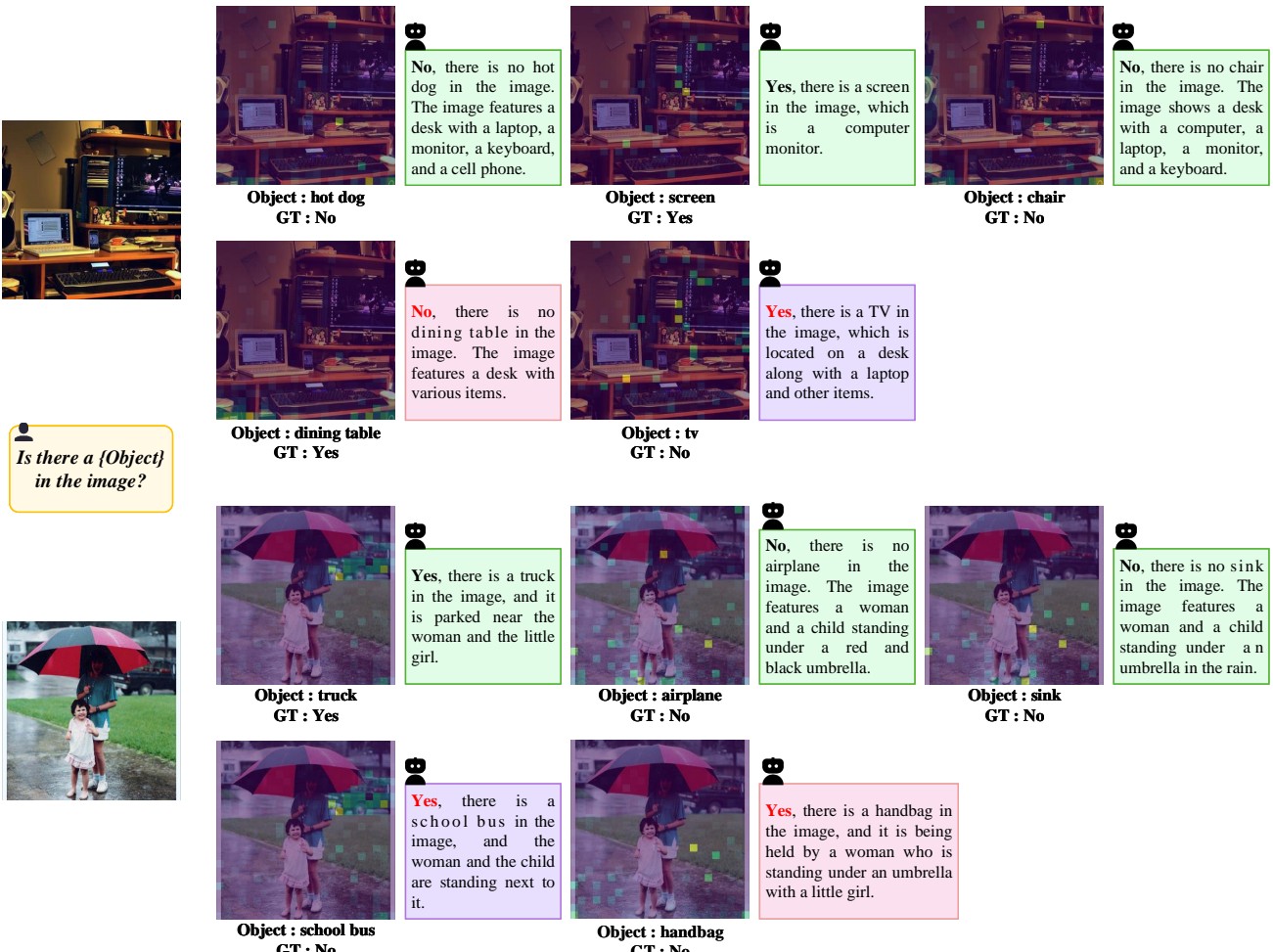

*Figure 7.* Visualization of different hallucination patterns. We asked multiple questions on the same sampled image to observe distinct hallucination patterns. In the figure, green boxes represent correct responses, purple boxes represent hallucinatory responses with PSD, and pink boxes represent hallucinatory responses with LF. Note: Some queries and labels are artificially designed and are intended solely to demonstrate the phenomenon of hallucination.

## A.2. Further Analysis in the Frequency Domain

We further analyze the frequency-domain characteristics of visual tokens across different LVLMs to demonstrate the broad compatibility of our FLASH with diverse models.

### A.2.1. VISION HEAD SELECTION STRATEGY

We provide additional visualizations of visual heads and other heads across various MHA layers of different LVLMs. These results are illustrated in Figures 8 and 9.

These findings demonstrate that the observed spectral characteristics are intrinsic to different functional heads rather than being artifacts of particular models or datasets. Consequently, leveraging spectral diagrams as a proxy for head selection is a

methodologically sound strategy.

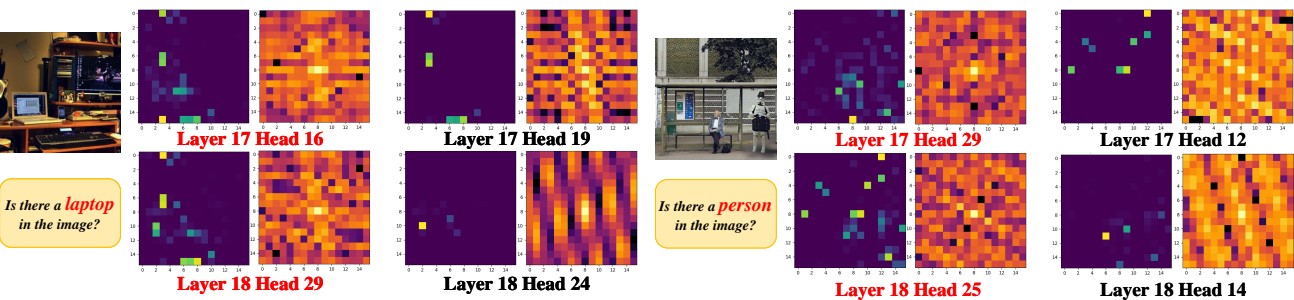

*Figure 8.* Visualization of Attention Heads in Shikra. Due to the intrinsic spatial resolution of the 256-token encoding, we recommend viewing these visualizations at a reduced scale to better perceive the emergent patterns and structural features. In the figure, red text indicates the vision head.

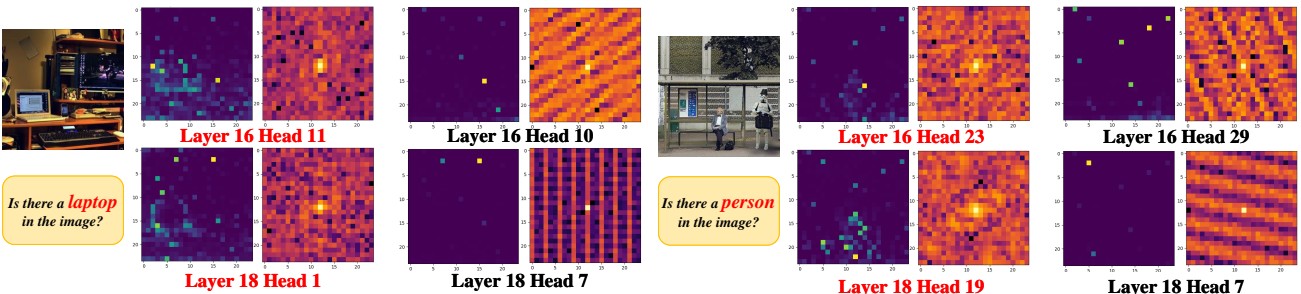

*Figure 9.* Visualization of Attention Heads in LLaVA-1.5. In the figure, red text indicates the vision head.

### A.2.2. SPECTRAL ANALYSIS OF INFORMATION FLOW WITHIN MHA

In Figure 3, we present only the curves for threshold $= 0.5$. (The matching curve for queries and keys is similar.) In this section, we provide additional curve results for different threshold values, shown in Figure 10.

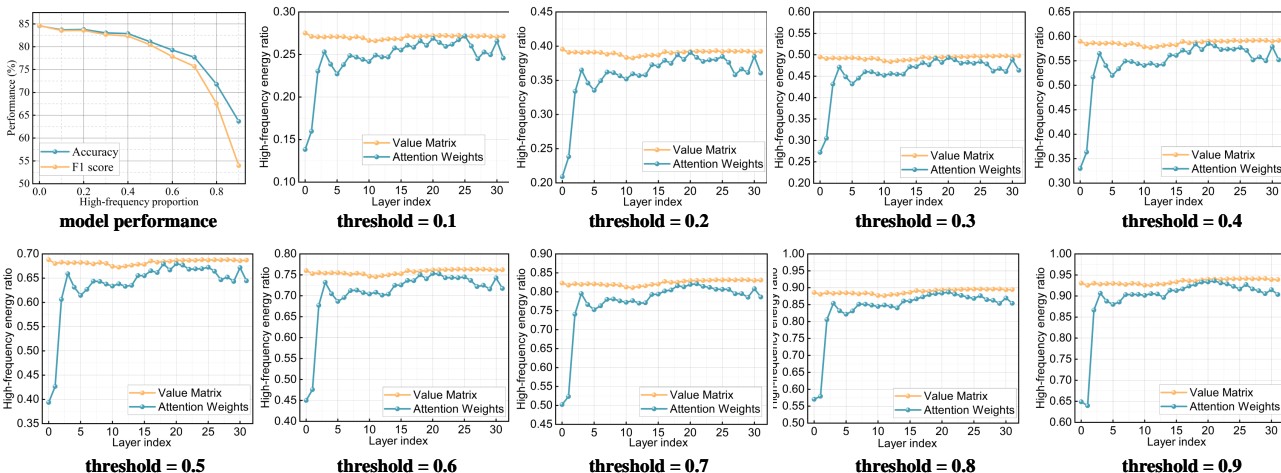

*Figure 10.* Proportion of spectral energy in value and attention outputs across varying high-frequency thresholds. Thresholds represent the top percentile of frequency components (e.g., 0.1 denotes the top 10%). When conducting statistical experiments, we employ random sampling methods.

As illustrated in Figure 10, for thresholds $< 0.4$, the model effectively restores high-frequency energy proportions in deep layers (e.g., maintaining levels up to layer 25 when threshold $= 0.1$ and layer 20 when threshold $= 0.2$). Consequently, the model exhibits robust performance within this lower threshold regime. However, as the threshold exceeds 0.4, this restorative capacity diminishes, leading to a precipitous decline in overall performance.

### A.3. Theoretical Explanation

In this section, we introduce the rationale for modulating V-stream in logarithmic frequency spectrum and how energy across different frequency bands in our DSM is adaptively transferred. These theorems correspond to Section 4.3 of the main text.

#### A.3.1. RATIONALE FOR LOG-MAGNITUDE DOMAIN MODULATION

In Section 4.3, we propose performing spectral intervention of V-stream in the log-magnitude domain as defined in Eq. 13. We provide a theoretical justification for this design:

Spectral coefficients of visual features (Value matrix) in Transformers typically follow a power-law distribution, where the magnitude $|C(f)|$ at frequency $f$ scales as $|C(f)| \propto f^{-\alpha}$ (Field & David, 1987; Simoncelli & Olshausen, 2001). This results in a massive dynamic range where low-frequency components are orders of magnitude larger than high-frequency ones. A linear modulation $\tilde{C} = \omega \cdot C$ would be highly sensitive to the choice of $\omega$; a small $\omega$ might be insufficient for high-frequency enhancement, while a slightly larger $\omega$ could lead to numerical overflow in low-frequency regions. By operating in the log-domain:

$$\log|\tilde{C}| = \mathcal{M}_v \cdot \log|C|, \tag{20}$$

the power-law relationship is linearized. This transformation is equivalent to a power-law refinement: $|\tilde{C}| = |C|^{\mathcal{M}_v}$. This allows the modulation mask $\mathcal{M}_v$ to act as an exponential scaling factor that adjusts the contrast between frequency bands rather than their absolute values, providing a much more stable control over the spectral slope.

Additionally, we conducted ablation experiments to compare the impact of logarithmic domain modulation and linear domain modulation on model performance. The relevant experimental results and analysis are presented in Section A.5.

#### A.3.2. THEORETICAL ANALYSIS OF ADAPTIVE ENERGY MODULATION

Eq. 15 introduces a Frobenius norm constraint to calibrate the modulated features. In this section, we prove that this functions as an adaptive energy reallocation mechanism through the lens of Parseval's Theorem.

**Energy Conservation in Frequency and Spatial Domains:** According to Parseval's Theorem for the Discrete Cosine Transform (DCT), the energy of a signal is preserved between the spatial and frequency domains:

$$||V||_F^2 = \sum_{i,j} |V|^2 = \sum_{u,v} |C_{u,v}|^2 = ||C||_F^2 \tag{21}$$

Therefore, the constraint in Eq. 15 can be interpreted as a normalization of total spectral energy:

$$||V^*||_F = ||V||_F \Rightarrow ||C^*||_F = ||C||_F \tag{22}$$

**The Zero-Sum Dynamics:** Let the total energy of the original spectrum be:

$$E_{total} = E_{low} + E_{high}. \tag{23}$$

When we enhance the high-frequency components in the log-domain, the new energy of the intermediate spectrum $\tilde{C}$ becomes:

$$E_{enhanced} = \tilde{E}_{low} + \tilde{E}_{high} \tag{24}$$

where $\tilde{E}_{high} > E_{high}$ (assuming $\mathcal{M}_v > 1$ for high frequencies).

By applying the normalization factor $\gamma = \frac{||V||_F}{||\tilde{V}||_F}$, the final energy distribution becomes:

$$E^* = \gamma^2 \tilde{E}_{low} + \gamma^2 \tilde{E}_{high} = E_{total} \tag{25}$$

Since $\tilde{E}_{high}$ has been increased significantly by the modulation, the denominator $||\tilde{V}||_F$ increases, resulting in $\gamma < 1$. Consequently, the term $\gamma^2 \tilde{E}_{low}$ is forced to decrease relative to the original $E_{low}$.

Therefore, Eq. 15 forces the model to pay for the high-frequency enhancement by automatically draining redundant energy from the low-frequency bands. This ensures that our modulation operates not by simple linear amplification, but by redistributing the model's attention within a fixed representation budget, thus maintaining numerical stability and pre-trained alignment. This proof also applies to the constraint process of S-stream.

## A.4. Performance of Different Tasks on the MME Dataset

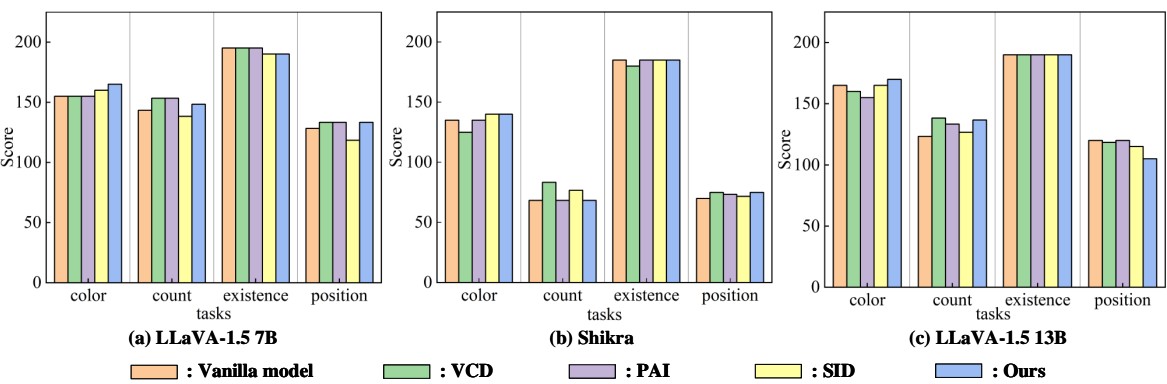

*Figure 11.* Performance of Different Tasks on the MME Dataset.

## A.5. Ablation Experiment and Parameter Sensitivity Analysis

Owing to space constraints, the main text focuses on core ablation studies. This section provides supplementary analysis and extended findings, including comprehensive parameter sensitivity tests (Table 3 and Figure 12) and a detailed evaluation of the logarithmic-domain modulation scheme (Table 4) referenced in Section 4.3. Additionally, we present the qualitative results of the task ablation experiments in Figure 13.

*Table 3.* Comparison of performance across different hyperparameters. In the table, orders 1–5 represent sensitivity experiments for $\lambda_v$; orders 6–10 represent sensitivity experiments for $\lambda_s$; orders 11–15 represent sensitivity experiments for $k$. The green shaded rows indicate the hyperparameter combinations used in this work.

| Orders | $\lambda_v$ | $\lambda_s$ | $k$ | POPE-Random | | | |
| --- | --- | --- | --- | --- | --- | --- | --- |
| | | | | Accuracy ↑ | F1 ↑ | Precision ↑ | Recall ↑ |
| Greedy | / | / | / | 88.97 | 88.90 | 89.41 | 88.40 |
| 1 | 1.0 | 0.9 | 5 | 89.63 | 89.37 | 91.72 | 87.13 |
| 2 | 1.2 | 0.9 | 5 | 90.03 | 89.77 | 92.20 | 87.47 |
| 3 | 1.4 | 0.9 | 5 | 89.97 | 89.69 | 92.25 | 87.27 |
| 4 | 1.6 | 0.9 | 5 | 89.83 | 89.52 | 92.41 | 86.80 |
| 5 | 1.8 | 0.9 | 5 | 89.77 | 89.42 | 92.52 | 86.53 |
| 6 | 1.2 | 0.9 | 5 | 90.03 | 89.77 | 92.20 | 87.47 |
| 7 | 1.2 | 0.7 | 5 | 90.03 | 89.76 | 92.26 | 87.40 |
| 8 | 1.2 | 0.5 | 5 | 90.03 | 89.76 | 92.26 | 87.40 |
| 9 | 1.2 | 0.3 | 5 | 90.03 | 89.76 | 92.26 | 87.40 |
| 10 | 1.2 | 0.1 | 5 | 90.00 | 89.73 | 92.19 | 87.40 |
| 11 | 1.2 | 0.9 | 1 | 89.83 | 89.57 | 91.93 | 87.33 |
| 12 | 1.2 | 0.9 | 3 | 89.97 | 89.71 | 92.07 | 87.47 |
| 13 | 1.2 | 0.9 | 5 | 90.03 | 89.77 | 92.20 | 87.47 |
| 14 | 1.2 | 0.9 | 8 | 89.97 | 89.71 | 92.07 | 87.47 |
| 15 | 1.2 | 0.9 | 10 | 89.97 | 89.71 | 92.07 | 87.47 |

## A.6. Comparison of Inference Performance and Efficiency

We demonstrate a comparison of the performance and efficiency of different methods in this section. We evaluated the inference performance of LLaVA-1.5 (Liu et al., 2024a) integrated with various hallucination mitigation strategies (Leng et al., 2023; Liu et al., 2024c; Huo et al., 2025), alongside their latency multipliers relative to the vanilla model. The result is shown in Table 5.

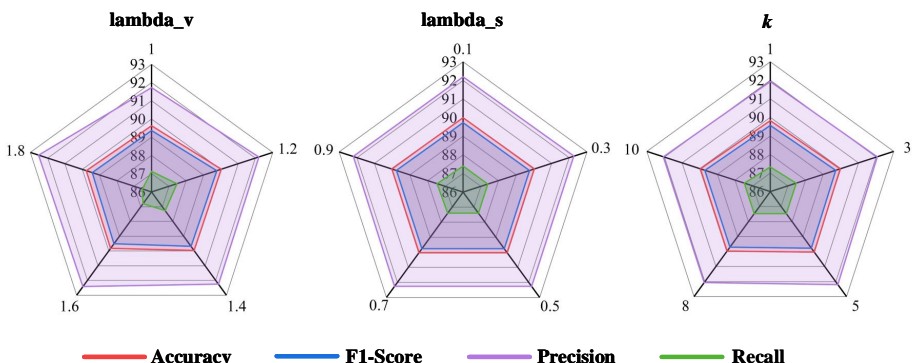

*Figure 12.* Comparison of parameter sensitivity experiment results for key hyperparameters.

*Table 4.* Comparison of modulation strategies in DSM. The **V.** and **S.** in the table represent V-stream and S-stream respectively. **Logarithmic** and **Linear** represent modulation of the spectrum in the logarithmic domain and linear domain, respectively.

| Orders | Modulation Space | | POPE-Random | | | |
|---|---|---|---|---|---|---|
| | Logarithmic | Linear | Accuracy ↑ | F1 ↑ | Precision ↑ | Recall ↑ |
| 1 | V. | S. | 90.03 | 89.77 | 92.20 | 87.47 |
| 2 | S. | V. | 89.77 | 89.51 | 91.80 | 87.33 |
| 3 | V. + S. | / | 90.00 | 89.74 | 92.13 | 87.47 |
| 4 | / | V. + S. | 89.70 | 89.43 | 91.85 | 87.13 |

*Table 5.* Comparison of performance and efficiency across different methods. In the table, **Average** represents the performance average across the three POPE-MSCOCO splits. **Times** reflects the multiple of the inference time of the baseline model (relative to the baseline method) when applying different methods on the POPE-bench dataset.

| Methods | POPE-R | | POPE-P | | POPE-A | | Average | | CHAIR | | Times |
|---|---|---|---|---|---|---|---|---|---|---|---|
| | Acc. ↑ | F1 ↑ | Acc. ↑ | F1 ↑ | Acc. ↑ | F1 ↑ | Acc. ↑ | F1 ↑ | $C_S$ ↓ | $C_I$ ↓ | |
| LLaVA-1.5 | 88.97 | 88.90 | 85.63 | 86.03 | 79.23 | 80.99 | 84.61 | 85.31 | 48.10 | 12.75 | 1× |
| +VCD | 89.07 | 88.99 | 85.60 | 85.99 | 79.27 | 81.00 | 84.65 | 85.33 | 48.80 | 12.85 | 1.98× |
| +PAI | 89.30 | 89.27 | 86.07 | 86.45 | 79.23 | 81.06 | 84.87 | 85.59 | 47.80 | 12.35 | 1.87× |
| +SID | 89.40 | 89.04 | 85.93 | 85.93 | 80.33 | 81.38 | 85.22 | 85.45 | 48.10 | 12.40 | 2.58× |
| +Ours | 90.03 | 89.77 | 86.53 | 86.62 | 80.53 | 81.75 | 85.70 | 86.05 | 47.50 | 12.55 | 1.69× |

## A.7. Qualitative Analysis of Generated Results

Due to space constraints in the main text, we present here a qualitative comparison of results from the generation task. The two images are sampled from the AMBER dataset (Wang et al., 2023).

## A.8. More Detailed Experimental Settings

In this section, we report more extensive experimental settings to supplement the relevant content in the section 5.1.

### A.8.1. DATASETS & EVALUATION METRICS

Following (Suo et al., 2025; Leng et al., 2023; Huo et al., 2025; Liu et al., 2024c; Wang et al., 2024b; Yin et al., 2025), we employ POPE (Li et al., 2023b) and MME (Fu et al., 2025) as benchmarks for discriminative tasks. For generative tasks, we select the widely recognized CHAIR (Rohrbach et al., 2019) and AMBER (Wang et al., 2023) benchmarks.

**POPE (Li et al., 2023b):** The Polling-based Object Probing Evaluation (POPE) is designed to evaluate hallucinations in LVLMs. It comprises three datasets: MSCOCO (Lin et al., 2015), A-OKVQA (Schwenk et al., 2022), and GQA (Hudson & Manning, 2019). Built upon the VQA task, POPE divides each dataset into three splits—*Random*, *Popular*, and *Adversarial*—totalling 27,000 queries. Each split samples 500 images, with six questions per image using the template: "***Is there a {object} in the image?***". The objects are sourced based on different criteria: the *Random* split selects objects

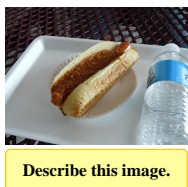

**Describe this image.**

**LLaVA-1.5 7B:**

The image features a delicious hot dog with chili and onions, placed on a paper plate. The hot dog is positioned in the center of the plate, and the plate is placed on a dining table. The table is surrounded by chairs, with one chair on the left side and another on the right side. In addition to the hot dog, there is a bottle of water located on the right side of the table, providing a refreshing beverage to accompany the meal.

**+ FLASH:**

The image features a dining table with a white tray holding a hot dog with chili and mustard. The hot dog is placed in the center of the tray, and the chili and mustard are spread evenly across the bun. The table also has a bottle of water positioned to the right side of the tray, completing the meal setup. The table appears to be made of metal, giving it a sturdy and durable appearance.

**+ FLASH (S-stream only):**

The image features a dining table with a white plate holding a hot dog with chili and mustard. The hot dog is placed in the center of the plate, and the chili and mustard are spread evenly across the top. The table also has a bottle of water positioned to the right side of the plate, ready to be enjoyed with the hot dog.

**+ FLASH (V-stream only):**

The image features a dining table with a white tray holding a hot dog with chili and mustard. The hot dog is placed on a bun, and the tray is positioned on the table. There is a bottle of water located next to the tray, providing a refreshing beverage to accompany the meal. The table appears to be a dining table, and the scene is set in a casual dining environment.

**+ FLASH (w/o head selection):**

The image features a dining table with a white plate holding a hot dog with chili and mustard. The hot dog is placed on a bun, and the plate is positioned on the table. There is a bottle of water located next to the plate, providing a refreshing beverage to accompany the meal. The table appears to be a dining table, and the scene is set in a casual dining environment.

*Figure 13.* Qualitative comparison of the ablation study results.

| | | | | |
|---|---|---|---|---|
| **Sample 1** | **Question: Describe this image.** | **LLaVA-1.5**: The image features a delicious hot dog with chili and onions, placed on a paper plate. The hot dog is positioned in the center of the plate, and the plate is placed on a dining table. The table is surrounded by chairs, with one chair on the left side and another on the right side. In addition to the hot dog, there is a bottle of water located on the right side of the table, providing a refreshing beverage to accompany the meal. | **LLaVA-1.5 + PAI**: The image features a hot dog on a paper plate, placed on a dining table. The hot dog is covered in chili and has a bun, making it a delicious-looking meal. The hot dog is accompanied by a bottle of water, which is placed next to the plate. The water bottle is located on the right side of the table, while the hot dog is on the left side. The table appears to be made of metal, giving it a sturdy and durable appearance. | |
| | | **LLaVA-1.5 + Ours**: The image features a dining table with a white tray holding a hot dog with chili and mustard. The hot dog is placed in the center of the tray, and the chili and mustard are spread evenly across the bun. The table also has a bottle of water positioned to the right side of the tray, completing the meal setup. The table appears to be made of metal, giving it a sturdy and durable appearance. | **LLaVA-1.5 + SID**: The image features a hot dog with chili on a bun, placed on a white paper plate. The hot dog is accompanied by a bottle of water, which is located to the right of the plate. The dining table occupies the majority of the scene, with the hot dog and the water bottle placed on it. The table appears to be made of metal, giving it a sturdy and durable appearance. | |
| | | **LLaVA-1.5 + VCD**: The image features a delicious hot dog with chili and onions, placed on a paper plate. The hot dog is positioned in the center of the plate, and the plate is placed on a dining table. The table is surrounded by chairs, with one chair on the left side and another on the right side. In addition to the hot dog, there is a bottle of water located on the right side of the table, providing a refreshing beverage to accompany the meal. | | |
| **Sample 2** | **Question: Describe this image.** | **LLaVA-1.5**: The image features a large white airplane parked on a runway at an airport. The airplane is positioned in the middle of the scene, with its nose pointing towards the left side of the image. In the background, there is a mountain visible, adding a scenic element to the image. Additionally, there are two people in the scene, one located near the left side of the image and the other closer to the right side. They might be airport staff or passengers waiting for their flight. | **LLaVA-1.5 + PAI**: The image features a large Delta Airlines airplane parked on the runway. The airplane is white and appears to be a large jetliner. It is parked on the runway, and the mountain range can be seen in the background. The airplane is parked on the runway, and it seems to be a large jetliner. | |
| | | **LLaVA-1.5 + Ours**: The image features a large white airplane parked on a runway, with a mountain in the background. The airplane is positioned on the left side of the scene, and its size is quite noticeable. The mountain in the background adds a sense of depth and scale to the image. | **LLaVA-1.5 + SID**: The image features a Delta Connection airplane parked on a runway, with a mountain in the background. The airplane is white and appears to be a small jetliner. It is situated on the tarmac, and there are a few people visible around the airplane, likely attending to maintenance or boarding. The scene captures the essence of an airport setting with the airplane as the main focus. | |
| | | **LLaVA-1.5 + VCD**: The image features a large white airplane parked on a runway at an airport. The airplane is positioned in the left of the scene, with its nose pointing towards the left side of the image. The airplane appears to be a commercial jet, possibly a Delta Airlines plane, as it is parked on the tarmac. In the background, there is a mountain visible, adding a scenic element to the image. Additionally, there are two people in the scene, one located near the left side of the image and the other closer to the right side. They might be airport staff or passengers waiting for their flight. | | |

*Figure 14.* Qualitative comparison of different methods for the generative task.

randomly from the dataset; the *Popular* split picks the most frequent objects; and the *Adversarial* split identifies objects that frequently co-occur with ground-truth objects but are absent, making it the most challenging. In this work, we use the

MS-COCO dataset (9,000 queries) as the representative testbed, employing Accuracy and F1-score as the evaluation metrics, formulated as:

$$\text{Accuracy} = \frac{TP + TN}{TP + FP + FN + TN}, \tag{26}$$

$$\text{F1-Score} = 2 \times \frac{(\text{Precision} \times \text{Recall})}{(\text{Precision} + \text{Recall})}, \tag{27}$$

where:

$$\text{Recall} = \frac{TP}{TP + FN}, \tag{28}$$

$$\text{Precision} = \frac{TP}{TP + FP}. \tag{29}$$

**MME (Fu et al., 2025):** MME is a comprehensive benchmark designed to evaluate the perceptual and cognitive faculties of LVLMs across 14 distinct subtasks. By employing a binary "Yes/No" response framework, it effectively minimizes the interference of models' linguistic inductive biases. While the perception track assesses fundamental attributes such as object existence, count, color, and position, the cognition track demands higher-level reasoning. In accordance with the protocol in (Cho et al., 2025; Leng et al., 2023), we evaluate different methods on four representative perception subtasks—**Existence, Count, Position, and Color**—and report the performance using the Score metrics (Fu et al., 2025).

**CHAIR (Rohrbach et al., 2019):** The Caption Hallucination Assessment with Image Relevance (CHAIR) has been widely adopted to evaluate hallucinations in generative tasks. As a captioning-based benchmark, it prompts LVLMs to describe input images in detail and calculates the proportion of mentioned objects that do not exist in the ground-truth label pool. CHAIR provides two granularities: instance-level ($\text{CHAIR}_I$) and sentence-level ($\text{CHAIR}_S$), formulated as:

$$\text{CHAIR}_I = \frac{|\{\text{hallucinated objects}\}|}{\text{all mentioned objects}}, \tag{30}$$

$$\text{CHAIR}_S = \frac{|\{\text{caption with hallucinated objects}\}|}{\text{all captions}}. \tag{31}$$

Following previous studies (Liu et al., 2024c; Zheng & Zhang, 2025), we randomly sample 500 images from MS-COCO (Lin et al., 2015) and use the prompt: "***Please help me describe the image in detail.***"

**AMBER (Wang et al., 2023):** AMBER is a recently developed multi-dimensional benchmark focused on hallucination evaluation in MLLMs. It offers fine-grained annotations and an automated, LLM-free evaluation pipeline, avoiding the high cost of GPT-4 APIs while ensuring efficiency and accuracy. Compared to CHAIR (Rohrbach et al., 2019), AMBER features a more extensive ground-truth label pool and covers existence, attributes, and relationships. We utilize the Generative task split (1,004 images, prompt: "***Describe this image.***") and follow the standard metrics: Cover, Hal, and Cog (Wang et al., 2023). We randomly selected 200 images from the collection for testing.

### A.8.2. BASELINES

To demonstrate the efficacy and generalization of FLASH, we integrate it into two representative LVLM families, covering diverse scales:

**LLaVA-1.5(Liu et al., 2024a):** As a prominent representative of end-to-end trained LVLMs, LLaVA-1.5 refines the original LLaVA framework by adopting a more powerful vision-language connector. It utilizes a CLIP-ViT-L/14 visual encoder and a Vicuna-v1.5 LLM, bridged by a two-layer MLP projection matrix. Despite its architectural simplicity, LLaVA-1.5 achieves SOTA performance on various academic benchmarks through the use of high-resolution image inputs and a diverse mix of instruction-tuning data. We select this model to validate FLASH's effectiveness on the standard MLP-based vision-language paradigm. In this paper, we selected the 7B and 13B version of LLaVA-1.5.

**Shikra(Chen et al., 2023a):** Shikra is designed to excel in "Referential Dialogue," a task requiring the model to precisely ground objects mentioned in the conversation. Unlike models that rely on auxiliary detection heads or specialized pre-training, Shikra introduces a unified framework that handles spatial coordinates (bounding boxes) as natural language tokens. This design allows it to perform seamless spatial reasoning and object localization within a multi-turn dialogue. By including Shikra, we aim to demonstrate that FLASH can mitigate hallucinations even in models with strong inherent grounding capabilities. In this paper, we selected version 7B.

A.8.3. IMPLEMENTATION DETAILS

For the compared method, we conducted testing using their official code. On the CHAIR dataset, we randomly sampled 500 images for inference using the query: "***Please help me describe this image in detail.***" For the AMBER dataset, we randomly sampled 200 images for inference testing, adhering to the official AMBER configuration (Wang et al., 2023) with the query: "***Describe this image.***" For both generation tasks, we report the average results of tests conducted under two different random seed sets. Additionally, beyond the experimental details reported in Section 5.1, we only modulate the first token for the discriminative task. The hyperparameter configurations for different models are shown in the Table 6.

*Table 6.* Hyperparameters for different models. In the table, "V/S-stream layer" indicates the layer within the MHA where the corresponding modulation scheme is applied.

| | $\lambda_s$ | $\lambda_v$ | S-stream Layer | V-stream-layer | $\tau$ | $k$ |
|---|---|---|---|---|---|---|
| **Discrimination Task** | | | | | | |
| LLaVA-1.5 7B | 0.9 | 1.2 | 16-27 | 3-32 | 0.2 | 5 |
| Shikra 7B | 0.9 | 1.1 | 16-27 | 3-32 | 0.4 | 5 |
| LLaVA-1.5 13B | 0.9 | 5.0 | 16-27 | 3-40 | 0.2 | 5 |
| **Generation Task** | | | | | | |
| LLaVA-1.5 7B | 0.1 | 1.2 | 16-27 | 3-32 | 0.2 | 5 |
| Shikra 7B | 0.9 | 1.1 | 16-27 | 3-32 | 0.4 | 5 |
| LLaVA-1.5 13B | 0.1 | 1.2 | 16-27 | 3-40 | 0.2 | 5 |

