# OpenReview forum: "Beyond Attention Imbalance: Mitigating Hallucinations via Spectral Surgery"
_ICML.cc/2026/Conference — ICML 2026 regular_

### Official Review · Reviewer_JdkD · 2026-02-23

**Soundness:** 3
**Presentation:** 3
**Significance:** 2
**Originality:** 3
**Overall Recommendation:** 4
**Confidence:** 3

**Summary:**

This paper identifies two patterns of hallucination: 'Perceptual-Semantic Dissociation' and 'Localized Fixation.' Building upon these findings, the authors propose the FLASH (Frequency-Localized Attention SHaping) algorithm. This method employs a Spectral Vortex Score to detect specific attention heads across layers and adaptively modulates the information flow during the decoding phase, effectively suppressing hallucinations.

**Compliance With Llm Reviewing Policy:**

Affirmed.

**Final Justification:**

My concerns have been fully addressed in the rebuttal, so I will raise my score.

**Key Questions For Authors:**

The empirical evaluation is currently limited to LLaVA-1.5. A significant concern is that the simple mistakes analyzed in this work rarely occur in today’s mainstream Large Vision-Language Models (LVLMs). Is the fundamental analysis of the hallucination problem in this paper built upon an issue that has largely been resolved in contemporary advanced open-source models? Could the authors provide more complex, realistic failure cases to demonstrate the ongoing relevance of these specific hallucination modes?

The experimental results lack comparisons with more advanced, recent mitigation algorithms (e.g., [MemVR, ICML], [EAH, EMNLP], [Vissink, ICLR], and [VHR, ACL]). Furthermore, the baseline models used for evaluation are relatively dated, with most published in 2023. Can the authors provide additional experimental results evaluating FLASH on newer visual models, whether they are general-purpose or domain-specific?

The proposed algorithm introduces three hyperparameters, but the manuscript does not provide a methodology for efficiently searching or tuning these parameters across different model architectures. Could the authors clarify if there is a systematic approach, heuristic, or adaptive strategy for fast hyperparameter search when deploying this algorithm on new architectures?

**Limitations:**

yes

**Strengths And Weaknesses:**

Strengths:

The paper provides a deeper analysis by further categorizing hallucinations induced by attention imbalance into two distinct modes: Perceptual-Semantic Dissociation (PSD) and Localized Fixation (LF).

Unlike previous approaches that simply rely on the magnitude of attention scores, this work proposes a novel metric, the Spectral Vortex Score (SVS), to effectively locate anomalous attention heads across different layers.
The proposed FLASH (Frequency-Localized Attention SHaping) algorithm is well-designed, enabling the model to successfully maintain a granular focus on localized regions while simultaneously increasing its focus on the broader global context.

Weakness:

As observed in Table 1, the empirical performance of the proposed algorithm is somewhat underwhelming. The results are largely comparable to the existing baselines, showing only minor improvements rather than significant leaps in performance.
The experimental evaluation is missing comparisons with several highly relevant, recent state-of-the-art methods for hallucination mitigation. To properly establish the effectiveness and superiority of FLASH, the authors should include comparisons with the following baselines:

•	[MemVR] Look Twice Before You Answer: Memory-Space Visual Retracing for Hallucination Mitigation. (ICML 2025)

•	[EAH] Seeing Clearly by Layer Two. (EMNLP 2025)

•	[Vissink] See What You Are Told: Visual Attention Sink in Large Multimodal Models. (ICLR 2025)

•	[VHR] Cracking the Code of Hallucination in LVLMs with Vision-aware Head Divergence. (ACL 2025)

The proposed approach heavily relies on manually chosen hyperparameters ($\lambda_v$, $\lambda_s$, and $k$). The paper currently lacks an adaptive strategy or robust guidelines for selecting these values across different models or downstream tasks, which raises concerns about the method's generalizability, robustness, and practical applicability out-of-the-box.

---

> ### Author Rebuttal · Authors · 2026-03-30
>
> Thank you for your valuable feedback and constructive comments. Additional **Figures and Tables in our responses** can be found at the $\color{red}{[Link]}$: https://anonymous.4open.science/r/FLASH_ICML/README.md
>
> **Summary**
> -
> In this paper, we demonstrate that recent post-hoc hallucination mitigation methods rely heavily on CD. By analyzing spatial attention and spectral features of visual tokens, we categorize hallucinations into PSD and LF patterns. We introduce **FLASH**, a training-free framework that mitigates hallucinations from a spectral perspective via two core components: vision head selection and dual-stream adaptive spectral modulation.
>
> **Responses to Comments**
> -
> **W1 & Q1: Concerns regarding experimental results, benchmarks, and comparisons.**
>
> **R1:**
>
> ***Regarding Experimental Performance:*** The results in the manuscript are obtained using **greedy decoding** (do_sample=False and beam=1). In this regime, overall gains are naturally less pronounced for all methods. However, under the same configuration, FLASH's improvements over SOTA remain competitive, suggesting that modest margins are attributable to the decoding regime rather than our method's limitations.
>
> For discriminative tasks, FLASH consistently achieves the best overall performance. For generative tasks, while some metrics show degradation, similar trends appear in SOTA methods. We attribute this to:
> 1) **Inherent Stochasticity in CHAIR:** Evaluation involves random sampling from COCO; despite using two random seeds (following prior work), fluctuations persist.
> 2) **Benchmark Reliability:** As noted in the Appendix, **AMBER** is a more advanced protocol than CHAIR. On AMBER, FLASH achieves superior results across all relevant metrics. We conducted experiments using updated benchmarks (MMMU-PRO) and evaluation methods (LLaVA-bench evaluation based on GPT4).  Results can be found in **Tabs. 8 & 9 at [Link]**.
>
> Therefore, after **considering all the performance evaluations**, we believe that FLASH has achieved comparable performance to the SOTA methods.
>
> ***Additional Evidence:*** To further validate FLASH, we conducted supplementary experiments (results at **[Link]**):
> 1. **Extensive Comparisons:** We compared FLASH against DoLA, VISTA, MFCD, VHR, and EAH using POPE-MSCOCO and AMBER (**Tabs. 1 & 2 at [Link]**). Notably, VISTA suffers from language collapse on AMBER, which we omitted. FLASH maintains a clear performance advantage under greedy decoding.
> 2. **Modern LVLMs (Qwen 3.5-9B (2026), LLaVA-NeXT-7B):** Visualization (**Fig. 4 & 5 at [Link]**) confirms these architectures still suffer from the two proposed hallucination patterns.
> 3. **Generalization:** Evaluations on MME-Bench and CHAIR (**Tabs. 6 & 7 at [Link]**) show FLASH effectively mitigates object hallucinations in SOTA LVLMs (Qwen 3.5-9B, LLaVA-NeXT-7B).
>
> These experimental results demonstrate that our FLASH not only is competitive among the SOTA methods, but also can help the more advanced LVLMs in the recent period to alleviate hallucinations.
>
> **W2 & Q2: Concerns about hyperparameter selection.**
>
> **R2:**
>
> **Sensitivity and Role of Hyperparameters:** We provide sensitivity analysis in Appendix A.5 (Tab. 3 in the manuscript), showing FLASH is robust to parameter variations within reasonable ranges. The three parameters serve distinct roles:
>
> 1) **$\lambda_s$ (S-stream strength):** Formulated as $\cal{M} = (1-\lambda_s) + \lambda_s \times softmask$, where $softmask$ suppresses high frequencies via DCT.
>    - **Task-dependency:** Generation tasks require stable visual injection, so we use a smaller $\lambda_s$ to maintain coherence. Discriminative tasks require stronger signals to separate categories, justifying a larger $\lambda_s$.
>    - **Resolution-dependency:** For models like Shikra-7B、Qwen 3.5-9B (smaller 2D maps, stronger spatial bias), natural diffusion in logits is prominent. We use a larger $\lambda_s$ here to ensure sufficient frequency modulation and stable visual signal transmission.
> 2) **$\lambda_v$ (V-stream strength):** This scales with model capacity. For smaller models (LLaVA-1.5-7B、Shikra-7B), we set $\lambda_v \in \\{1.1, 1.2\\}$. For LLaVA-1.5-13B, deeper layers and stronger representations allow a larger $\lambda_v$ to enhance frequency-domain structural signals, boosting both discriminative and generative performance.
> 3) **$k$ (Candidate visual heads):** We use a fixed empirical heuristic, setting $k \approx 15\\%$ of the total heads (e.g., $k=5$ for 32-head models like LLaVA; $k=2$ for 16-head models like Qwen 3.5).
>
> In summary, the hyperparameters are grounded in clear physical meanings and follow a principled heuristic, making FLASH highly practical for deployment on new LVLMs.
>
> We once again express our gratitude for your efforts on our manuscript. **If you have any further questions, please feel free to discuss them with us during the second round of discussion. If we resolve your concerns, please reconsider our rating score. Thanks!**

---

> > ### Author Rebuttal · Reviewer_JdkD · 2026-04-03
> >
> > Thanks for the detailed response. I've raised my score.

---

> > > ### Author Response · Authors · 2026-04-03
> > >
> > > We would like to once again thank you for your positive feedback on our manuscript and rebuttal. Best wishes!

---

### Official Review · Reviewer_bftr · 2026-03-09

**Soundness:** 3
**Presentation:** 2
**Significance:** 3
**Originality:** 3
**Overall Recommendation:** 4
**Confidence:** 4

**Summary:**

The goal of this paper is to alleviate hallucinations in Large Vision Language Models. The paper critiques prior works that employ attention reweighting and Contrastive Decoding to emphasize visual features while de-emphasizing the influence of language priors. The authors claim that attention reweighting is coarse, and such spatial manipulations do not ``fix" the underlying problems in how LVLM processes vision. They provide a spectral analysis of the MHA layers and identify two causes for hallucinations: (1) Perceptual-Semantic Dissociation (PSD) and Localized Fixation (LF). The authors identify that the primary driver of hallucinations is the smoothening (low-pass filtering) effect of the MHA mechanism. The High-Frequency (HF) energy deficit between attention weights and Values, and between queries and keys, drives PSD and LF, respectively.

The proposed training-free framework, FLASH, uses spectral redistribution to reduce the HF deficit. FLASH introduces the Spectral Vortex Score (SVS) and Dual-Stream Spectrum Modulation (DMS). SVS identifies vision heads in MHA, ensuring that only the relevant features pertaining to vision are altered without compromising language features. The DMS operates on the heads chosen by SVS. DMS-V-stream mitigates PSD by modulating value matrices in the DCT frequency domain and pushing energy from low-frequencies to high-frequencies using logarithmic modulation. DMS-S-stream handles LF by modulating attention scores and pushing energy from high-frequencies to low-frequencies using linear modulation.

**Compliance With Llm Reviewing Policy:**

Affirmed.

**Final Justification:**

The authors' responses have addressed my concerns about limited baselines, limited task evaluations, and missing clarifications.

**Key Questions For Authors:**

See Weaknesses

**Limitations:**

Yes, the authors have provided limitations.

**Strengths And Weaknesses:**

**Strengths**:

1. The spectral analysis of hallucinations and the proposed spectral modulation offer a deeper diagnosis and solution to the hallucination problem in contrast to current training-free methods using spatial manipulations like attention reweighting.

2. Replacing Contrastive Decoding offers significant latency improvements.

3. Hallucination intervention by zero-sum energy redistribution is theoretically sound and the justifications for targeting attention scores and logarithmic value modulation are coherent and make intuitive sense.

4. Results in Table 1 exhibit FLASH's considerable performance gains on discriminative tasks over baselines.

**Weaknesses**:

*Major Concerns*:

1. The authors only provide two possible hallucination scenarios: PSD and LF, but it is unclear whether these encompass all hallucinations in LVLMs. Since the proposed solution FLASH directly operates on the failure points of PSD and LF, there seems to be no scope for generalization and a lack of discussion on whether FLASH's interventions degrade the LVLM performance on other hallucination types.

2. Lack of technical clarity in Section 3, particularly the definition of High-frequency (HF) energy ratio and the corresponding threshold. How is the HF ratio computed? Is it using a radial distance following Eqn. 11?

3. Limited baseline comparisons. The paper restricts comparisons to PAI, VCD, and SID. Methods discussed in related works like OPERA, DoLA, ICD and other CD-based methods are missing for a comprehensive evaluation of FLASH. Moreover, like FLASH that hypothesizes HF deficit for hallucinations, authors of [1] propose a CD solution based on the same hypothesis is relevant for comparison. VISTA [2] which performs visual steering is also missing from discussion.

4. Limited task evaluation. Evaluations on LLM-guided benchmarks/metrics are missing such as (1) long-form open-ended generation benchmarks: LLaVa-bench using GPT-4V judge (2) SHR metric in HA-DPO that would account for other hallucination types like relational, positional, etc. These benchmarks/metrics would highlight if spectral energy redistribution performs well for long generations and its interaction with varied hallucinations.

5. Figures 6 and 8 show the spatial and spectral visualization of vision and language heads. Corresponding visualizations before and after FLASH application and before and after CD-based method application are missing.

6. There is no explanation for using two metrics in SVS: Spectral Concentration and Spectral Isotropy. Ablations on using just one are missing. The two metrics together perhaps offer a very restrictive selection strategy that might not have cross-model applicability.

7. Comparisons with improvements over LLaVA-1.5 such as LLaVA-NeXT specialized in high resolution and other models like Qwen-VL are missing. Q-former based models are also lacking but the authors have expressed this in limitations.

8. Ablation in Table 4 does not offer strong confirmation for using logarithmic modulation for V-stream and linear for S-stream. Orders 1 and 3 are almost identical and despite this S-stream is linearly modulated. Perhaps comparisons on different dataset or model is needed.

9. Performance improvement of FLASH is negligible or not significant especially in Table 11.

*Minor Questions*:

1. Section 3.3 claims that MHA acts as a low-pass filter and that the attention output fails to recover the richness of the Value matrix. Why then is the modulation applied to the Value matrix (V-stream) rather than the Attention weights/Scores to counteract this filtering effect?

2. What dataset and model was used to plot Figures 3 and 4? and are the pink and purple lines overlapping in Figure 4a?

3. Where is citation or explanation of claim in Ln. 285: "leveraging its superior energy compaction properties..."

4. Why is generation performance poorer compared to discriminative task in Table 1? There is no explanation offered.

5. In Table 2, do the last two rows include SVS selection or not?

6. What are the challenges to applying FLASH to Q-former models? The limitations are vague on this.

7. How were the V and S stream layers chosen in Table 6? Based on plots 3b and 4b?

8. Which dataset and model was used in Tables 3,4 and Figure 12?

9. In Figure 10, the plots are very similar. The explanation offered in Appendix 2.2.2 talks about significant decline for thresholds beyond 0.4 which is not clear.

10. This is very minor but I do not see a dining table in Figure 7 but the ground truth is yes.


[1] Liu, Bingqian, et al. "Multi-Frequency Contrastive Decoding: Alleviating Hallucinations for Large Vision-Language Models." Proceedings of the 2025 Conference on Empirical Methods in Natural Language Processing. 2025.

[2] Li, Z., Shi, H., Gao, Y., Liu, D., Wang, Z., Chen, Y., Liu, T., Zhao, L., Wang, H. and Metaxas, D.N.. (2025). The Hidden Life of Tokens: Reducing Hallucination of Large Vision-Language Models Via Visual Information Steering. Proceedings of Machine Learning Research, 2025.

---

> ### Author Rebuttal · Authors · 2026-03-30
>
> Thank you for your valuable feedback. Additional **Figures/Tables are at: [link]** https://anonymous.4open.science/r/FLASH_ICML/README.md. Due to space, please refer to our responses to other reviewers for similar comments.
>
> **Summary**
> -
> In this paper, we demonstrate that recent post-hoc hallucination mitigation methods rely heavily on CD. By analyzing spatial attention and spectral features of visual tokens, we categorize hallucinations into PSD and LF patterns. We introduce **FLASH**, a training-free framework that mitigates hallucinations from a spectral perspective via two core components: vision head selection and dual-stream adaptive spectral modulation.
>
> **Responses** (Ma: Major / Mi: Minor)
> -
> **Ma.1: Object Hallucinations Focus.** Following previous works, our study focuses on object hallucinations. While various hallucination forms exist, we concentrate on PSD and LF as our research objective. We will clarify this scope and strengthen “hallucination type” descriptions in the next version. Future work will explore FLASH’s impact on broader hallucination patterns.
>
> **Ma.2: High-Frequency Ratio.** Defined as the proportion of high-pass filtered energy to total spectral energy: (1) Compute total energy; (2) Apply high-pass filters with varied cutoff frequencies; (3) Measure high-frequency energy; (4) Ratio = (3)/(1). The threshold (e.g., 0.1) denotes the top % of retained spectral components.
>
> **Ma.3, 4, 7, 9, and Mi.4: Additional Results and Performances.** Expanded results are in **Tabs. 1-2, 6-9 [Link]**. See **R1 (Reviewer JdkD) and R2 (Reviewer 6utG)** for details.
>
> **Ma.5: Visualization of Modulation Results.** CD-based methods operate at the logit level and do not involve MHA head modulation, making direct qualitative head comparisons inapplicable. We provide spatial/spectral visualizations and heatmaps before/after FLASH modulation in **Figs. 2-3 [Link]**. Changes in the Value matrix (representing raw image info) are less intuitively observable than the Score matrix, hence we emphasize the former.
>
> **Ma.6: SVS Components & Generalization.** Spectral concentration and isotropy are complementary. As shown in Fig. 6 in the manuscript, unconcentrated low-frequency energy disperses into higher frequencies, aligning with our insights (L242-264). FLASH’s applicability across LLaVA-1.5 (7B/13B) and Shikra (Figs. 6, 8, 9 in the manuscript) proves it is not family-specific but generalizes to projector-based LVLMs. Component ablations are in **Tab. 3 [Link]**.
>
> **Ma.8: Modulation Domain Ablation.** To validate our scheme, we performed ablations using Shikra on MME (**Table 5 [Link]**). Results confirm that selecting the appropriate modulation domain (Log. vs. Line.) is crucial for performance gains.
>
> **Mi.1: Score vs. Value Modulation.** The Value matrix encodes raw image info, but LF hallucinations stem from attention fixation on irrelevant regions. Merely strengthening representations (V-stream) is insufficient; we must disperse the model’s attention via the Score matrix modulation to redirect the model’s focus.
>
> **Mi.2: Fig. 3 & 4 Details.** Generated using LLaVA-1.5-7B on POPE-MSCOCO Random split. In Fig. 4a, overlapping lines occur because different queries on the same image share identical visual sequence inputs to the key projector, resulting in the same spectral distribution.
>
> **Mi.3: Typos.** We thank the reviewer. We corrected “energy compression” to “energy concentration,” citing [1] regarding DCT’s properties.
>
> [1] N. Ahmed, T. Natarajan and K. R. Rao. Discrete Cosine Transform. IEEE Transactions on Computers. doi: 10.1109/T-C.1974.223784.
>
> **Mi.5: SVS Selection.** All experiments utilized SVS-based selection. Sub-item ablations are provided in **Tab. 3 [Link]**.
>
> **Mi.6: Limitations of Q-Former-based Methods.** FLASH requires spatial consistency for 1D-to-2D spectral transformation. Q-Former disrupts this structure, preventing direct application. However, we are exploring adaptations for Q-Former architectures and will present these in future work.
>
> **Mi.7: Layer Selection.** We selected the modulation layers based on the empirical roles of MHA (Shallow: reconstruction; Mid: fusion; Late: lexical mapping) and Figs. 3b & 4b. This pattern is consistent across different LVLM families of the same scale.
>
> **Mi.8: Setup of Tab. 3, 4 and Fig. 12 .** Experiments used LLaVA-1.5-7B on the POPE-MSCOCO Random split.
>
> **Mi.9: Threshold Analysis.** "Significant decline" denotes performances drops (see Fig. 10-a). The subsequent figures in Fig. 10 explain this phenomenon. At threshold ≤0.4, HF energy alignment between attention and Value enables close observation. Above 0.4, this capability drops. Visual similarity at 0.9 is an artifact of excessive information masking, leading to a random state.
>
> **If you have further questions, please feel free to discuss them with us during the second round of discussion. If we resolve your concerns, please reconsider our rating score. Thanks!**

---

> > ### Author Rebuttal · Reviewer_bftr · 2026-04-02
> >
> > I have read the authors' rebuttal and the other reviewers' comments. The author's rebuttal resolved the questions raised, and I have no further questions. I will maintain my positive rating for this paper.

---

> > > ### Author Response · Authors · 2026-04-03
> > >
> > > Thank you once again for your thoughtful review. We are greatly encouraged by your insightful question, as it demonstrates a grasp of the central concepts behind our work. We appreciate the time and effort you have dedicated to reviewing our paper.

---

### Official Review · Reviewer_8poS · 2026-03-14

**Soundness:** 2
**Presentation:** 2
**Significance:** 3
**Originality:** 3
**Overall Recommendation:** 3
**Confidence:** 4

**Summary:**

This paper challenges the prevailing assumption that hallucinations in LVLMs stem primarily from cross-modal attention imbalance. The authors first demonstrate empirically that performance gains from existing methods largely derive from Contrastive Decoding (CD) rather than attention re-weighting itself. They then propose a finer-grained taxonomy of hallucinations, identifying two distinct failure modes: Perceptual-Semantic Dissociation (PSD), where the model localizes the correct region but fails at semantic discrimination, and Localized Fixation (LF), where attention pathologically fixates on irrelevant visual regions. Both modes are interpreted through a frequency-domain lens. Building on this analysis, the authors propose FLASH, a training-free and CD-free framework that introduces the Spectral Vortex Score (SVS) to identify vision-specific attention heads, and applies a dual-stream spectral modulation (DSM) to adaptively reshape the frequency profiles of value matrices and attention scores within MHA layers. Experiments across POPE, MME, CHAIR, and AMBER benchmarks show that FLASH achieves competitive or superior performance compared to CD-based baselines with lower inference overhead.

**Compliance With Llm Reviewing Policy:**

Affirmed.

**Key Questions For Authors:**

1. PSD and LF are defined through qualitative attention map inspection at a single MHA layer. Can you provide a quantitative criterion for classifying samples into each type, and report per-type performance breakdowns showing that V-stream and S-stream interventions differentially benefit PSD and LF samples respectively? This is central to validating the paper's core thesis.

**Limitations:**

The authors discuss limitations in Appendix A.9, noting the remaining computational cost from spectral transforms and untested compatibility with Q-Former-based architectures. However, the discussion omits several important concerns: the practical implications of per-model hyperparameter dependence, the mechanism behind inconsistent generative performance, and the operationalizability of the PSD/LF taxonomy. A more thorough limitations discussion would strengthen the paper's credibility.

**Strengths And Weaknesses:**

**Strengths:**
1. The core diagnostic observation is genuinely useful to the community. The finding that spatial attention re-weighting alone provides minimal gains and that prior methods' effectiveness is largely attributable to CD is a concrete and well-supported claim (Fig. 2). This reframes how future work should evaluate and credit individual components in hallucination mitigation pipelines.

2. The appendix covers parameter sensitivity and modulation strategy comparisons reasonably thoroughly.

**Weakness:**
1. PSD/LF lack quantitative definitions; there's no way to classify a sample at inference time, which undermines the dual-stream motivation.

2. Key SVS choices (R=1, equal-weight summation) are unjustified and unablated.

3. Generative results are inconsistent (Shikra CHAIR_I degrades) with no mechanistic explanation.

---

> ### Author Rebuttal · Authors · 2026-03-30
>
> Thank you for your valuable feedback. **Additional Figures/Tables** are at **[Link]**: https://anonymous.4open.science/r/FLASH_ICML/README.md
>
> **Summary**
> -
> In this paper, we demonstrate that recent post-hoc hallucination mitigation methods rely heavily on CD. By analyzing spatial attention and spectral features of visual tokens, we categorize hallucinations into PSD and LF patterns. We introduce FLASH, a training-free framework that mitigates hallucinations from a spectral perspective via two core components: vision head selection and dual-stream adaptive spectral modulation.
>
> **Responses**
> -
> **W1 & Q1:** PSD/LF lack quantitative definitions; there's no way to classify a sample at inference time, which undermines the dual-stream motivation.
>
> **R1: Explicit classification is not the optimal design choice for FLASH.** We initially explored a strategy to identify PSD and LF patterns by calculating the Entropy of the Score matrix and applying targeted modulations. While this yielded marginal performance gains, it **introduced computational overhead and inference latency, which contradicts the core design philosophy of FLASH.**
>
> Instead, we adopted a unified modulation strategy (applying Score and Value modulation to all samples), achieving a better balance. Our reasoning is as follows:
> 1) Robustness against PSD (Seeing but Misidentifying): In PSD hallucinations, the model incorrectly identifies objects within regions sharing coarse-grained similarities with the ground truth, resulting in disproportionately high attention scores on these tokens.  Although using S-stream slightly disperses attention, the inherent focus remains anchored to key tokens.
> 2) Resilience against LF (Irrelevant Local Focus): Applying V-stream modulation does not amplify errors because the preceding S-stream has already redistributed attention (Fig. 7 in the manuscript), prevent irrelevant areas of the value matrix from being enhanced.
> 3) Empirical Evidence: Visualizations of spatial/frequency spectrums before and after modulation (**Figs. 2-3 [Link]**) confirm that unified modulation consistently refines focus across both hallucination types.
>
> Therefore, the transition from explicit classification to unified modulation is a logical optimization. It delivers a superior performance-to-speed ratio, ensuring FLASH remains a potent solution for LVLM deployments. **The dual-stream synergy naturally handles the variance between PSD and LF patterns, proving that a specialized classifier is redundant and computationally inefficient.** We also added ablation studies (**Table 10 [Link]**) to provide evidence.
>
> **W2:** Key SVS choices (R=1, equal-weight summation) are unjustified and unablated.
>
> **R2:** We believe this stems from a misunderstanding of our technical design.
>
> **Circular Mask & R=1:** Unlike rectangular masks, we employ a circular mask in the frequency domain for smoother low-frequency extraction, a standard practice in signal processing. We set $R=1$ to isolate the extreme low-frequency (DC-centered) components for precise $\cal{L}_r$ calculation. For low-resolution feature maps, this produces a cross-pattern mask centered on the DC component (**Fig. 1 [Link]**).
>
> **Equal-Weight Summation:** The SVS (Eq. 9) is defined by the sum of $\cal{L}_r$ and $\cal{O}_c$. Since both are bounded within [0,1] and represent equally contributions, we utilize equal weighting to avoid introducing unnecessary hyperparameters.
>
> We added a fine-grained ablation study on SVS and sensitivity analysis for $R$ in **Tables 3 & 4 [Link]**, which confirm that $R=1$ and equal weights provide the most robust performance across varied scenarios.
>
> **W3:** Generative results are inconsistent (Shikra CHAIR_I degrades) with no mechanistic explanation.
>
> **R3:** The results in the manuscript are obtained using greedy decoding (do_sample=False, beam=1), where gains for all post-hoc methods are naturally constrained. Yet, FLASH’s improvements over SOTA remain competitive. CHAIR exhibits inherent stochasticity due to random COCO sampling. Degradations in specific generative metrics are observed across competing SOTA methods, suggesting a shared sensitivity to the decoding regime rather than a flaw in FLASH. As noted in the Appendix, AMBER is a more comprehensive protocol. On AMBER, FLASH achieves superior results across all metrics. On discriminative tasks, FLASH consistently achieves SOTA overall.
>
> See **Tables 1,2, and 6–9 [Link]** for additional robust results.
>
> **Limitations:** Hyperparameters, generative performance, and PSD/LF practicality.
>
> **Responses:** Due to space limits: 1) See Reviewer JdkD (R2) for hyperparameter analysis. 2) See R3 for generative performance. 3) See R1 for PSD/LF practicality.
>
> We once again express our gratitude for your efforts on our manuscript. **If you have any further questions, please feel free to discuss them with us during the second round of discussion. If we resolve your concerns, please reconsider our rating score. Thanks!**

---

### Official Review · Reviewer_6utG · 2026-03-18

**Soundness:** 3
**Presentation:** 3
**Significance:** 3
**Originality:** 3
**Overall Recommendation:** 4
**Confidence:** 2

**Summary:**

This paper introduces two LVLMs' hallucination patterns, including perceptual-semantic dissociation and localized fixation. Then, it proposes a training-free and contrastive decoding-free framework, named FLASH, for addressing these two hallucination patterns. The key idea is using the spectral vortex score to select vision head and adjust spectral modulation to rectify visual information flow during decoding. Beyond its efficiency, it outperforms existing methods. I think the main benefit is superior efficiency.

**Compliance With Llm Reviewing Policy:**

Affirmed.

**Final Justification:**

The responses have addressed my concerns about validated models and benchmarks.

**Key Questions For Authors:**

I will change my rating according to the response to Weaknesses 1.

**Limitations:**

yes

**Strengths And Weaknesses:**

Strengths
1. Efficiency. It is the main strength of the proposed method. It is training-free and contrastive decoding-free.
2. It outperforms existing methods, including VCD, PAI, SID.
3. The motivation and presentation are clear.

Weaknesses
1. My main concern is that the validated models are old (llava 1.5 and Shikra). Recent lvlms (e.g., qwen 3.5) have improved the hallucination problem. Are the proposed hallucination patterns still significant?
2. Why the proposed method can improve MME (a general VQA benchmark) score? It is unclear. Also, I suggest to do evaluation on more recent benchmarks, e.g., HLE and MMMU-Pro.

---

> ### Author Rebuttal · Authors · 2026-03-30
>
> Thank you for your valuable feedback and constructive comments.
>
> **Suplementary Visualization Figures and Tables**
> -
> Additional **Figures and Tables in our responses** can be found at $\color{red}{[Link]}$: https://anonymous.4open.science/r/FLASH_ICML/README.md
>
> **Summary**
> -
> In this paper, we demonstrate that recent post-hoc hallucination mitigation methods rely heavily on CD. By analyzing spatial attention and spectral features of visual tokens, we categorize hallucinations into **PSD** and **LF** patterns. We introduce **FLASH**, a training-free framework that mitigates hallucinations from a spectral perspective via two core components: vision head selection and dual-stream adaptive spectral modulation.
>
> **Responses to Comments**
> -
> **W1:** My main concern is that the validated models are old (llava 1.5 and Shikra). Recent lvlms (e.g., qwen 3.5) have improved the hallucination problem. Are the proposed hallucination patterns still significant?
>
> **R1:** Thank you for your comments. The supplementary experiments can be divided into two parts:
>
> ***1. The proposed PSD and LF hallucination patterns are indeed persistent across more advanced LVLMs.*** To demonstrate this, we conducted supplementary experiments on **LLaVA-NeXT-7B (2024)** and **Qwen 3.5-9B (2026)**. Qualitative results and visualizations are provided in **Fig. 4 (LLaVA-NeXT-7B) and Fig. 5 (Qwen 3.5-9B) at $\color{red}{[Link]}$**. These results reveal that while newer architectures effectively mitigate hallucinations, they remain inherently susceptible to PSD and LF patterns. This underscores the necessity of our study.
>
> ***2. More Performance Evidence of our FLASH.*** In order to present more comprehensive experimental results, we expanded our evaluation to include comparisons with SOTA training-free methods and integration tests. Specifically:
>
> 1) We integrated FLASH into **LLaVA-NeXT-7B and Qwen 3.5-9B**, with results detailed in **Tables 6 and 7 at $\color{red}{[Link]}$**.
>
> 2) We **compared our FLASH with SOTA training-free-based methods**, including DoLA[1], VISTA[2], MFCD[3], VHR[4], and EAH[5]. The results can be found in **Tables 1 and 2 at $\color{red}{[Link]}$**.
>
> The empirical evidence shows that FLASH not only remains competitive against recent training-free methods but also enhances the hallucination-resistance of advanced LVLMs without requiring additional training.
>
> **W2:** Why the proposed method can improve MME (a general VQA benchmark) score? It is unclear. Also, I suggest to do evaluation on more recent benchmarks, e.g., HLE and MMMU-Pro.
>
> **R2:** Thank you for your comments.
>
> ***Responses to the Comments of MME-Bench.*** As described in lines 950-955 of the manuscript, MME is a widely adopted benchmark that evaluates LVLMs by transforming visual questions into a binary (Yes/No) format, similar to POPE. The improvement in MME scores is a direct consequence of FLASH’s architectural design. Specifically:
> 1)    FLASH expands the receptive field of the LVLM through the S-stream;
> 2)    FLASH enhances the model's ability to capture fine-grained object features through the V-stream;
> 3)    FLASH applys precise modulation through the SVS for vision head select.
>
> Since MME specifically evaluates the existence, color, position, and count of objects, which tasks indeed need above abilities, it is logically consistent that FLASH’s mechanism yields higher MME scores. We utilize the official MME scoring protocol to ensure fair comparison.
>
> ***More Benchmark Experiments.*** Following your suggestion, we conducted additional evaluations on LLaVA-Bench and the MMMU-Pro benchmark. The results, summarized in **Tables 8 and 9 at $\color{red}{[Link]}$**, demonstrate that FLASH consistently mitigates hallucinations across these more challenging benchmarks and more advanced evaluation strategies. These results further validate the robustness and generalization capability of our proposed method in complex reasoning and visual perception tasks.
>
> We once again express our gratitude for your efforts on our manuscript. **If you have any further questions, please feel free to discuss them with us during the second round of discussion. If we resolve your concerns, please reconsider our rating score. Thanks!**
>
> [1] DoLa: Decoding by Contrasting Layers Improves Factuality in Large Language Models (ICLR, 2024)
>
> [2] The Hidden Life of Tokens: Reducing Hallucination of Large Vision-Language Models Via Visual Information Steering (ICML, 2025)
>
> [3] Multi-Frequency Contrastive Decoding: Alleviating Hallucinations for Large Vision-Language Models (EMNLP, 2025)
>
> [4] Cracking the Code of Hallucination in LVLMs with Vision-aware Head Divergence (ACL, 2025)
>
> [5] Seeing Clearly by Layer Two: Enhancing Attention Heads to Alleviate Hallucination in LVLMs (EMNLP 2025)

---

> > ### Author Rebuttal · Reviewer_6utG · 2026-04-04
> >
> > well address my concerns about models and benchmarks.

---

> > > ### Author Response · Authors · 2026-04-04
> > >
> > > Thank you again for your necessary comments. Best wishes!

---

### Decision · Program_Chairs · 2026-04-30

**Decision:**

Accept (regular)

**Comment:**

The paper introduces a spectral approach to mitigating hallucinations in LVLMs and presents a training-free framework with solid empirical validation. During the rebuttal, the authors effectively addressed concerns by adding experiments on newer models, expanding benchmarks, and clarifying key design choices. As a result, 3 out of 4 reviewers raised their scores and acknowledged that their concerns were resolved, while the remaining reviewer did not update their rating despite the responses. Overall, the reviews converge toward a positive assessment of the paper’s quality and contribution.